# Formation Control of a Multi-Autonomous Underwater Vehicle Event-Triggered Mechanism Based on the Hungarian Algorithm

Juan Li [ID], Yanxin Zhang and Wenbo Li *

School of Intelligent Science and Engineering, Harbin Engineering University, Harbin 150009, China; lijuan041@hrbeu.edu.cn (J.L.); zhanyanxin@hrbeu.edu.cn (Y.Z.)
* Correspondence: liwenbo049@hrbeu.edu.cn

**Abstract:** Among the key technologies of Autonomous Underwater Vehicle (AUV) leader–follower formations control, formation reconfiguration technology is one of the main technologies to ensure that multiple AUVs successfully complete their tasks in a complex operating environment. The biggest drawback of the leader–follower formations technology is the failure of the leader and the excessive communication pressure of the leader. Aiming at the problem of leader failure in multi-AUV leader–follower formations, the Hungarian algorithm is used to reconstruct the failed formation with a minimum cost, and the improvement of the Hungarian algorithm can solve the problem of a non-standard assignment. In order to solve the problem of an increased leader communication task after formation reconfiguration, the application of an event-triggered mechanism (ETM) can reduce unnecessary and useless communication, while the efficiency of the ETM can be improved through increasing the event-triggered conditions of the sampling error threshold. The simulation results of multi-AUV formation control show that the Hungarian algorithm proposed in this paper can deal with the leader failure in the multi-AUV leader–follower formation, and the ETM designed in this paper can reduce about 90% of the communication traffic of the formation which also proves the highly efficient performance of the improved ETM in the paper.

**Keywords:** autonomous underwater vehicle; hungarian algorithm; formation reconfiguration; event-triggered mechanism



## 1. Introduction

At present, the technology of AUVs has been gradually maturing. The technology becomes outstanding in marine equipment combined with sensors, intelligent control technology, and communication technology, which is widely used in civil and military activities. Additionally, it is the main tool for performing tasks such as marine resource exploration, port reconnaissance, underwater demining, and laying pipelines on the seabed [1]. With the rapid development of information technology, the operation environment of AUVs will become more complex and difficult in the future. A single AUV often cannot complete complicated tasks due to its limited resources and bad system fault tolerance. The coordinated formation operation of a multi-AUV can make up for the limited operation capacity of a single AUV. Therefore, it is bound to become a trend to use large-scale, low-cost and multi-functional AUVs to form clusters to complete formation operations in the future.

The form of the leader–follower is a kind of formation. The basic idea is to select an AUV as the leader in the formation, with the other AUVs as followers. The advantage of this is that the control structure is simple and precise formation control can be realized. The leader is the core part of the formation and when the AUV formation performs tasks, the followers need to communicate with the leader constantly to confirm the position of the leader and to maintain the formation. The failure of the leader refers to the fact that the leader cannot continue sailing or communicating with the follower due to damage, causing the entire formation to become paralyzed. Therefore, the leader–follower formation can have problems such as leader failure and leader communication pressure.

In order to solve the problem of leader failure, some scholars have proposed to create a virtual leader to replace the failed leader. The literature [2] uses the planned trajectory of the virtual leader to plan the trajectory of each follower directly. In this case, since the leader is virtual, there is no formation problem due to leader failure. Liao R. et al. [3] designed a multi-agent system as a formation composed of a virtual leader with a completely unknown input and several real leaders and followers. Then, distributed finite-time observers were designed for each real leader and follower to obtain state estimates of the adjacent agents. Lalish and Morgansen et al. [4] combined a virtual navigator with a virtual structure, designated the desired movement path of each follower with the help of the virtual navigator, and finally completed the 2D plane formation control of the desired formation; however, the virtual leader method requires precise modeling of the leader model, which increases the complexity of the operation. The key to formation reconfiguration is how to reconstruct the formation from the decision-making level of the formation control. In 1995, Kuhn [5] suggested using the Hungarian algorithm to solve the optimal assignment problem for the first time. The Hungarian algorithm can deal with these kinds of assignment problems efficiently and is widely used in various fields, such as firepower-target allocation, resource distribution and the optimal allocation of computer low-end threads. Wang et al. [6] combined the Hungarian algorithm with the formation control algorithm which can deal with the dynamic allocation of robots in the formation process. The Hungarian algorithm was applied to the optimal assignment problem of cloud computing in [7], where the cost matrix was constructed according to the tasks to be calculated and the available cloud computing nodes, and then the Hungarian algorithm was used to solve the optimal assignment strategy of the cost matrix; however, the cost matrix in the literature was given, which was not obtained by a real-time settlement of each robot in the formation process or its own motion state.

Another disadvantage of the leader–follower formation algorithm is that the leader needs to communicate with its followers frequently, but most of the communication has little effect on the control of multi-agents. Moreover, it increases the workload of the system processor. In response to this problem, some scholars have proposed the method of an event-triggering mechanism. In 2009, Dimarogonas and Johansson [8] proposed to apply the event-triggered mechanism to the controlling application of multi-agents, which solved the problem of excessive communication pressure of the multi-agent system with output saturation. In order to solve the problem of agents needing to communicate continuously to meet the convergence requirements, Sariff N. et al. [9] adopted a synchronous perturbation random algorithm to integrate the event-triggering system into the design of a broadcast distributed, consistent linear controller. Zhang H. et al. [10] studied the leader–following consensus problem for a class of nonlinear multiagent systems. The novel event-triggered and asynchronous edge-event-triggered mechanisms were designed for the leader and all edges, respectively. The static and dynamic consensus protocols under these mechanisms were proposed to address the leader–following consensus problem for MASs with Lipschitz dynamics, and the systems did not exhibit Zeno behavior under these two control schemes. Lin Y. et al. [11] proposed a distributed event-triggering mechanism to achieve affine formation control, which could be implemented in an asynchronous manner and guarantee Zeno-free behavior. He N. et al. [12] studied the problems of asymptotic stability and queue stability in an autonomous fleet and used event-trigger control technology to overcome the problem of frequent acceleration/deceleration with a fixed cycle control, reducing the loss of vehicle formation, and improving the efficiency of the autonomous fleet control. Huang Hongwei of Southwest Jiaotong University [13] used an event-triggered mechanism for the consistent control of multi-agents. He improved the event-triggered mechanism and reduced the updating frequency of the system control input. Astrom [14] and Arzen [15] et al. designed a control strategy based on an event-triggering mechanism, so that the system can autonomously perform sampling and control updates according to the needs of the control task. Liu et al. [16] studied the event-triggered control problem of uncertain nonlinear systems with actuator faults. The actuator failures may be unknown,

therefore the total number of failures may be unlimited. In order to reduce the communication burden between the controller and the actuator, a new event-triggered control law was designed. Through analysis by Lyapunov, it was proved that the control protocol could ensure that all signals of the closed-loop system were globally bounded, and that the tracking errors of the system outputs could converge exponentially to an arbitrary small residual.

In summary, in order to solve the above two major problems, this thesis reconstructs a failed formation with minimum cost through the Hungarian algorithm which can deal with the leader failure problem in a multi-AUV leader–follower formation. For the problem of the leader communication task volume after a formation reconfiguration, the event-triggered mechanism is used to reduce unnecessary and useless communication. In addition, it proposes to improve the efficiency of an event-triggered mechanism by increasing the sampling error threshold. The flow chart of the control decision is shown below in Figure 1.

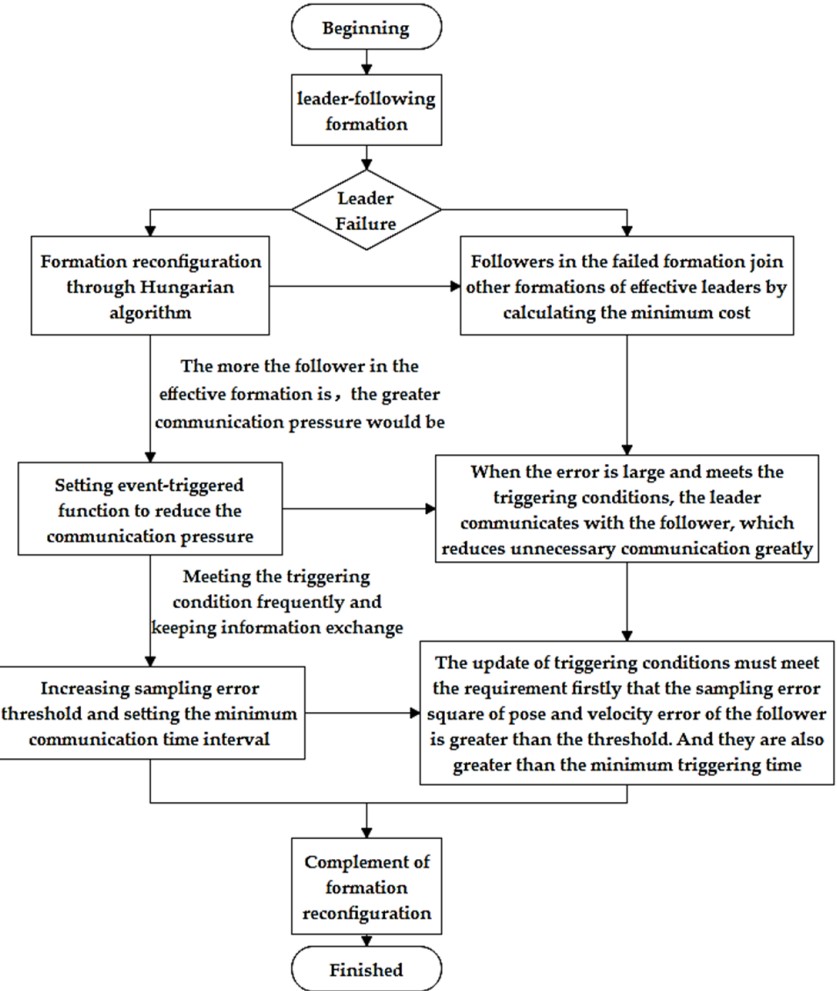

**Figure 1.** Control decision flow chart.

## 2. Construction of AUV Feedback Linearization Model

*2.1. AUV Nonlinear Model*

In Figure 2, $E - \xi\eta\zeta$ is the northeast coordinate system, i.e., the fixed coordinate system, and $O - xyz$ is the hull coordinate system, i.e., the inertial coordinate system.

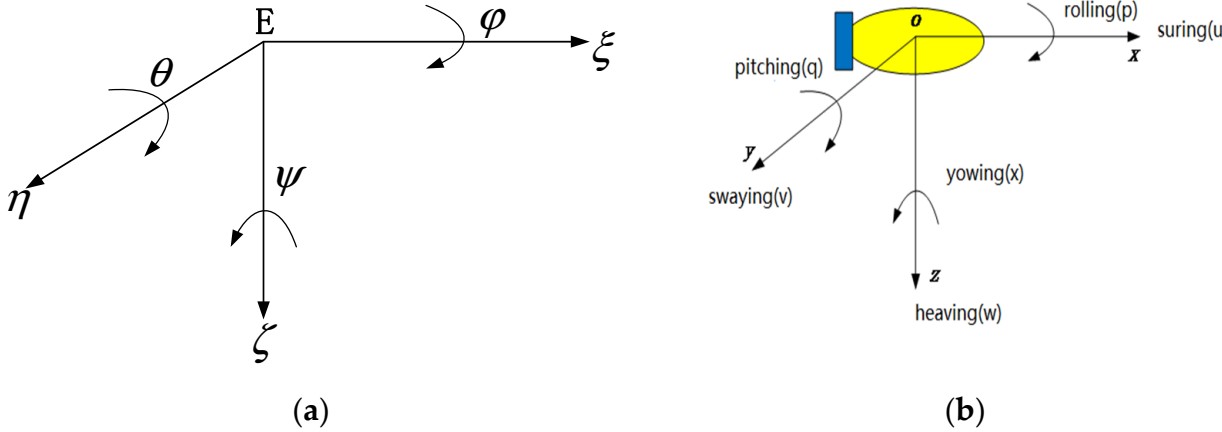

**Figure 2.** AUV coordinate diagram. (**a**) Inertial coordinate system. (**b**) Shipboard coordinate system.

The AUV is full-actuated. The following dynamics and kinematics model is developed in the body coordinate system:

$$\begin{cases} \dot{\eta} = J(\eta)v \\ M\dot{v} = g'\tau - C(v)v - D(v)v - g(\eta) \end{cases} \tag{1}$$

In the model: $\eta = (x,y,z,\theta,\psi)^T \in R^5$ is the AUV's position vector of the fixed coordinate system. The velocity vector of the AUV in the hull coordinate system is $v = (u,v,w,q,r)^T \in R^5$. $M$ is the inertial matrix and $J(\eta)$ is the transition matrix. $C(v)$ is the Coriolis and centripetal force matrices. $D(v)$ is the lifting force moment and hydrodynamic drag. The restoring force and torque vector is $g(\eta)$. The input vector of the AUV actuator is $\tau = (\tau_u, \tau_v, \tau_w, \tau_q, \tau_r)^T \in R^5$ and $g' \in R^{5\times5}$ is the parameter matrix of the actuator. The higher order damping term and the effect of hull swing on the AUV motion are not considered. The detailed mathematical model of the AUV is built as follows:

Kinematic model:

$$\begin{cases} \dot{x} = u\cos\psi\cos\theta - v\sin\psi + w\cos\psi\sin\theta \\ \dot{y} = u\sin\psi\cos\theta + v\cos\psi + w\sin\psi\sin\theta \\ \dot{z} = -u\sin\theta + w\cos\theta \\ \dot{\theta} = q \\ \dot{\psi} = r/\cos\theta \end{cases} \tag{2}$$

*2.2. The Feedback Linearization Model of the AUV*

The AUV model is appropriately transformed for the subject in this paper:

$$\begin{cases} \dot{\eta} = J(\eta)v \\ \dot{v} = M^{-1}g'\tau + M^{-1}N(\eta,v) \end{cases} \tag{3}$$

The above Equation (3) is transformed into the following form for the model linearization:

$$\dot{\xi} = f(\xi) + M_1 g'\tau \tag{4}$$

The output quantity of the nonlinear system is the position vector, and the nonlinear model of the AUV is as follows:

$$\begin{cases} \dot{\xi} = f(\xi) + g(\xi)\tau \\ \varsigma = h(\xi) \end{cases} \tag{5}$$

In the model: $f(\xi) = M_1 \begin{bmatrix} J(\eta)v \\ N(\eta, v) \end{bmatrix}, h(\xi) = (x, y, z, \theta, \psi)^T, g(\xi) = M_1 g'$.

It can be seen from the definition of the Lie derivative in literature [17]:

$$\dot\varsigma = \frac{\partial h}{\partial \xi}(f(\xi) + g(\xi)\tau)$$

The first Lie derivative $L_f h(\xi)$, $L_g h(\xi)$ can be obtained with the following form:

$$\begin{cases} L_f h(\xi) = J(\eta)v \neq 0 \\ L_g h(\xi) = 0 \end{cases}$$

Similarly, according to the definition of the second-order Lie derivative, we can obtain:

$$\ddot\varsigma = \frac{\partial L_f h(\xi)}{\partial \xi}(f(\xi) + g(\xi)\tau)$$

While the second-order Lie derivative has the following form:

$$\begin{cases} L_f^2 h(\xi) = \frac{\partial L_f h(\xi)}{\partial \xi} f(\xi) \neq 0 \\ L_g L_f h(\xi) = \frac{\partial L_f h(\xi)}{\partial \xi} g(\xi) \neq 0 \end{cases}$$

According to the definition in the literature [17], the sum of the relative orders of the model is: $\rho_1 + \rho_2 + \rho_3 + \rho_4 + \rho_5 = 10$, where $\rho_1 = \rho_2 = \rho_3 = \rho_4 = \rho_5 = 2$, that is, the sum of the relative order is equal to the order of the system 10. It can be seen that the AUV model can be accurately feedback linearized and has a solution:

Select coordinate changes as follows:

$$\begin{cases} z_1(\xi) = (h_1(\xi), h_2(\xi), h_3(\xi), h_4(\xi), h_5(\xi))^T \\ z_2(\xi) = (L_f h_1(\xi), L_f h_2(\xi), L_f h_3(\xi), \\ \qquad\quad L_f h_4(\xi), L_f h_5(\xi))^T \end{cases} \tag{6}$$

and

$$\begin{cases} \dot z_1 = z_2 \\ \dot z_2 = L_f^2 h(\xi) + L_g L_f h(\xi)\tau \end{cases} \tag{7}$$

If $U = L_f^2 h(\xi) + L_g L_f h(\xi)\tau$, then the actual control input under a new coordinate transformation is as follows: $\tau = (L_g L_f h(\xi))^{-1}(U - L_f^2 h(\xi))$.

The AUV linearized mathematical model of AUV is obtained as:

$$\begin{cases} \dot z_1 = z_2 \\ \dot z_2 = U \end{cases} \tag{8}$$

At this time, the construction of the AUV linearized model is achieved.

## 3. Hungarian Algorithm

In recent years, more and more scholars have begun to use advanced optimization algorithms to solve challenging practical problems from a decision-making level. When optimizing many-objective problems, HZA C. [18] et al. proposed a learning-based algorithm to match the characteristics of the problem. MA Dulebenets [19] proposed a new adaptive polyploid memetic algorithm to solve the scheduling problem of CDT trucks. At the same time, many problem-specific hybridization techniques have been used in the algorithm to facilitate the search process. ZZ Liu [20] et al. proposed a new algorithm, AnD, based on an angle-based selection strategy and a displacement-based density estimation strategy, which can delete poor individuals one by one in environmental selection. Junayed Pasha [21] et al. proposed a mixed integer linear programming model to solve

the "factory-in-a-box" planning problem. F Sherwani [22] et al. analyzed deterministic optimization techniques that can be effectively used in machine learning applications.

This manuscript mainly studies leader failure in the leader–follower formation. The leader–follower formation is a kind of central-station control. Once the leader in the formation fails, the entire formation system will become paralyzed, therefore, this paper uses the Hungarian algorithm to solve this problem at the decision-making level.

Inspired by the literature [23], the reconfiguration of a failed formation is regarded as an optimal assignment problem. In the failed formation, if the followers are in the communicating range of all leaders, the cost of assigning followers to the normal leader formation can be calculated by designing an appropriate loss function (i.e., cost function). There are three situations during the process of assignment: the followers are more than, less than, or equal to the normal leaders. The traditional Hungarian algorithm can solve the situation where the followers are equal to the normal leaders. For other special situations, the traditional Hungarian algorithm needs to be improved.

In the process of the multi-AUVs formation operation, the leader $F_l$ has M followers. During formation operating, if the leader fails, then the follower needs to be reconstructed to the other $n$ leader formations. The cost of the formation reconfiguration between the follower $i$ and the other leader is $C_i = [\cos ti1, \dots, \cos tin]$, respectively.

**Definition 1.** *If there are m followers and n leaders during formation reconfiguration, the cost will be at a minimum under the following condition:*

$$\cos t = \min \sum_{i=1}^{m} C_i$$

*where* $\cos t$ *is the total cost to complete this formation reconfiguration.*

### 3.1. Construction of Cost Model

Usually, there are four aspects for building the cost model: the path cost of leader–follower formation reconfiguration, communication cost and loss, and the additional cost.

1. Building the model of path cost

Supposing that the formation fails, the position of follower $i$ is $X_i = [x_i, y_i, z_i]$, the velocity is $v_i$, and the position of the effective leader $j$ is $X_j = [x_j, y_j, z_j]$. Then the path cost is:

$$\tau_{1ij} = p_1 E - p_2 v_i + p_3 l_{ij} \tag{9}$$

The unit of cost is the energy consumed. $E$ is the remaining energy of the AUV, and the remaining energy assumed in the paper is same. If $E = 60$, then the speed of $v_i$ is equal to follower $i$. $l_{ij}$ is the distance between follower $i$ and leader $j$, and $p_1, p_2, p_3$ are the coefficient weight values. The Equation (9) shows that the closer the distance between the follower and effective leader is, the higher the speed is, and therefore the smaller the path cost will be when reconstructing the formation.

2. Building the model of the communicating cost:

$$\tau_{2ij} = \begin{cases} \frac{l_{ij}}{R_j} \exp(-\frac{(x_i-x_j)^2}{\alpha^2} - \frac{(y_i-y_j)^2}{\beta^2} \\ \quad - \frac{(z_i-z_j)^2}{\gamma^2}) & l \leq R_j \\ +\infty & l > R_j \end{cases} \tag{10}$$

In the model: $R_j$ is the maximum broadcasting distance of the effective leader $j$. Once the follower $i$ is not within the communicating range of $j$, the communicating cost will be infinite. The closer the distance between follower $i$ and effective leader $j$ is, the smaller the communication cost will be. The coefficient weight value is $\alpha^2 = 300$, $\beta^2 = 120$, $\gamma^2 = 100$.

3. Communicating propagation loss model:

$$H = (D/4)^{\frac{1}{2}} \tag{11}$$

In the model: $D$ is the water depth of the AUV, and $H$ is the depth factor. If $l_{ij} < H$, this means that the distance from the receiver to the sender is less than the depth parameter, and the propagation loss $TL$ is [24]:

$$TL = 20\log(l_{ij}) + \alpha l_{ij} + 60 - k_L \tag{12}$$

where $\alpha$ is the absorption coefficient, and $k_L$ is the abnormal propagation decibels:

$$\alpha = c_2 \frac{f_2^2 f^2}{f_2^2 + f^2} + (1 - c_1 D)[c_3 \frac{f_3^2 f^2}{f_3^2 + f^2} + c_4 \frac{f^2}{f_3^2}] \tag{13}$$

In the model: $c_1 = 6.32 \times 10^{-5}$, $c_2 = \frac{0.54}{c} \times 10^{(0.69pH-6)}$, $c_3 = 2.03 \times 10^{-2} \cdot S$, $c_4 = 2.93 \times 10^{-2}$, $f_2 = 6.1(S/35)^{0.5} \cdot 10^{[3-\frac{1051}{T+273}]}(kHz)$, $f_3 = 21.9\exp[13.82 - \frac{3500}{T+273}](kHz)$.

$D$ is the water depth and $C$ is the sound velocity (km/S). $S$ is the salinity (%) and $pH$ is the $pH$ value of the environment. $T$ is degree Celsius (°) and $f$ is the frequency of the underwater acoustic communication. Generally, the transmission of the underwater acoustic communication is broadband signals, and signals transmit the frequency averagely.

$$f = \sqrt{f_a f_h} \tag{14}$$

where $f_a$ is the maximum frequency and $f_h$ is the minimum frequency of the signal transmission. If $H < l_{ij} < 8H$, the propagation loss $TL$ is:

$$TL = 15\log_{10}^{l_{ij}} + \alpha l_{ij} + \alpha_T(l_{ij}/H - 1) + 5\log_{10}^{l_{ij}} + 60 - k_L \tag{15}$$

In the formula: $\alpha_T$ is the shallow water absorption coefficient.
If $l_{ij} > 8H$

$$TL = 10\log_{10}^{l_{ij}} + \alpha l_{ij} + \alpha_T(l_{ij}/H - 1) + 10\log_{10}^{l_{ij}} + 64.5 - k_L \tag{16}$$

The cost model of communication loss is:

$$\tau_{3ij} = TL \tag{17}$$

The unit of transmission loss is decibel.

4. Additional model

Due to the influence of the service life on the leader and the follower in the formation process, the performance of communication and load will be different. Therefore, an additional cost model should be added in the formation reconfiguration. The additional cost model is designed by the designer according to the performance of each AUV, that is, $C_e \in R^{n \times m}$.

Through decomposition and normalization, the cost model of reassigning failure followers to effective leaders can be expressed by the cost matrix $C$:

$$C = \begin{bmatrix} C_{11} & C_{21} & \cdots & C_{m1} \\ C_{12} & C_{22} & \cdots & C_{m2} \\ \vdots & \vdots & \ddots & \vdots \\ C_{1n} & C_{2n} & \cdots & C_{mn} \end{bmatrix} + C_e \tag{18}$$

The value of the cost matrix $C$ can be calculated by the following formula:

$$\begin{cases} C_{ij} = \alpha_1 \times \tau_{1ij} + \alpha_2 \times \tau_{2ij} + \alpha_3 \times \tau_{3ij} \\ \alpha_1 + \alpha_2 + \alpha_3 = 1 \end{cases} \tag{19}$$

where $\alpha_1, \alpha_2, \alpha_3$ are the weight values of each sub-cost model, respectively.

### 3.2. The Improvement of the Hungarian Algorithm

When the assignment is performed by using the Hungarian algorithm, we should find the minimum value of each row and column of the matrix, then specify the row and column of the matrix to make "0" appear in each row and column of the matrix. Finally, the assignment scheme is determined according to the position where "0 "appears. If the follower is unequal to the effective leaders, this means there are $m$ followers and $n$ effective leaders ($m \neq n$). When the followers are less than the effective leaders ($m < n$), we need to design $n - m$ followers, and the corresponding $n - m$ zero elements are added to the cost matrix. On the contrary, ($m > n$), $m - n$ virtual leaders are added to the formation, which means the additional $m - n$ rows of zero elements are added in the cost matrix. For the assignment result, if the follower is assigned to the virtual leader, their cost information will continue to follow the above steps until the assignment of all followers is completed (attention: the added zero element in the cost matrix has no meaning). The execution steps of the algorithm are shown in Figure 3.

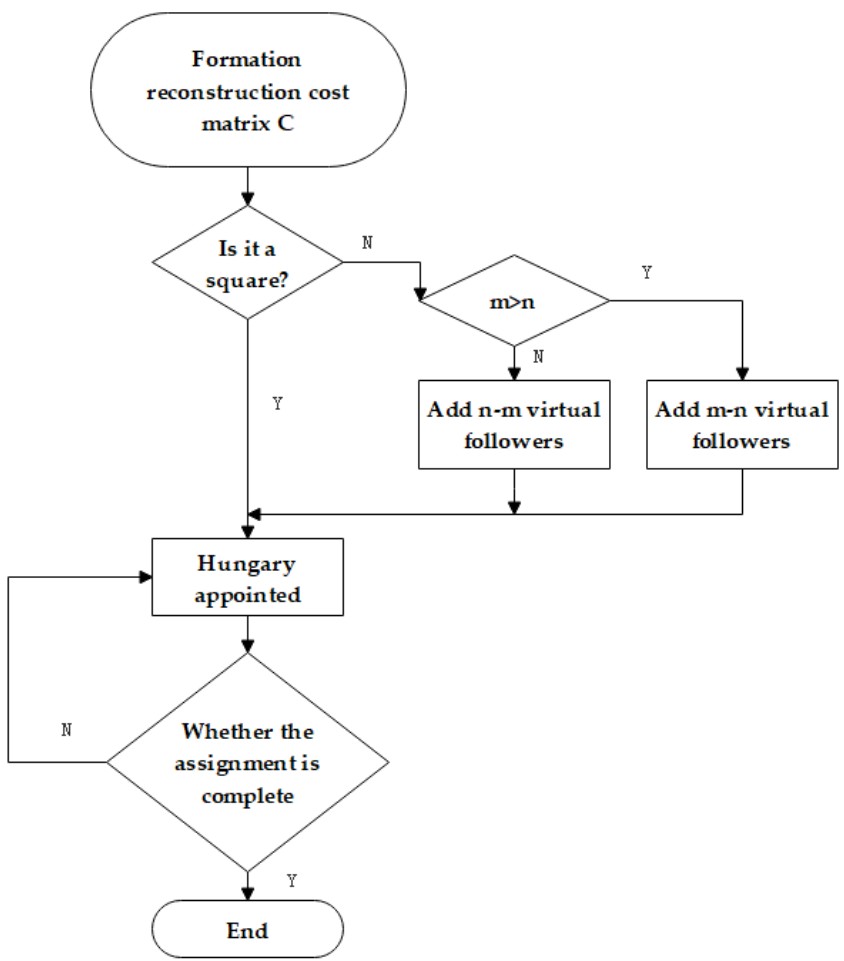

**Figure 3.** Flowchart of the improved Hungarian algorithm.

## 4. Formation Control under the Multi-AUV Event-Triggered Mechanism

### *4.1. The Design of AUV Formation Controller*

Formation requirements: The triangle/wedge formation can be realized based on the leader, and the formation can be maintained in a three-dimensional space. The formation is shown in Figure 4.

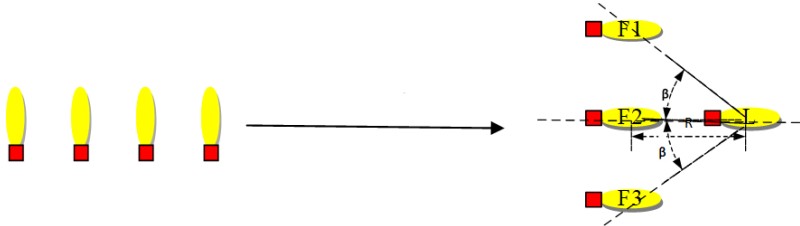

**Figure 4.** Schematic diagram of formation.

In Figure 4, L represents the leader, and F1, F2 and F3 are the followers of the AUV, respectively. If the formation is constrained by the constraint quantity in the schematic diagram, the formation cannot be maintained through a traditional constraint mode in a three-dimensional space. This paper proposes the following constraint method:

$$
\begin{cases}
\eta_L + d_1 = \eta_{F1} \\
\eta_L + d_2 = \eta_{F2} \\
\eta_L + d_3 = \eta_{F3} \\
\dot{\eta}_L + dv_1 = \dot{\eta}_{F1} \\
\dot{\eta}_L + dv_2 = \dot{\eta}_{F2} \\
\dot{\eta}_L + dv_3 = \dot{\eta}_{F3}
\end{cases}
\tag{20}
$$

R is the distance between the leader and the follower F2, and E is the attitude angle maintained by the formation. $\eta_L$ is the leader and $\eta_{Fi}$ is the motion state vector between the leader and follower *i*:

$$
\begin{cases}
d_1 = \left((\cos\beta)^{-1}R\cos(\psi_L - \beta - \pi), -(\cos\beta)^{-1}R\cos(\psi_L + \beta - \pi), 0, 0, 0\right)^T \\
d_2 = (R\cos(\psi_L - \pi), R\sin(\psi_L - \pi), 0, 0, 0)^T \\
d_3 = \left((\cos\beta)^{-1}R\cos(\psi_L + \beta - \pi), (\cos\beta)^{-1}R\cos(\psi_L - \beta - \pi), 0, 0, 0\right)^T \\
dv_1 = J_\eta(r_L\tan(\beta)R, 0, 0, 0, 0)^T \\
dv_2 = (0, 0, 0, 0, 0)^T \\
dv_3 = J_\eta(-r_L\tan(\beta)R, 0, 0, 0, 0)^T
\end{cases}
\tag{21}
$$

**Definition 2.** *If there are n AUVs in the leader–follower formation, and the motion state vector of the follower i at time t is $\varepsilon_i(t) = \eta_i(t)$, then the motion state vector of the leader is $\varepsilon_L(t) = \eta_L(t)$. If Equation (22) is satisfied, the formation maintenance and stability convergence can be realized:*

$$
\begin{aligned}
\lim_{t\to\infty}\left|\varepsilon_i(t) - \varepsilon_L(t) + d_i\right| &= 0 \\
\lim_{t\to\infty}\left|\dot{\varepsilon}_i(t) - \dot{\varepsilon}_L(t) + dv_i\right| &= 0
\end{aligned}
\qquad i = 1, 2, \ldots, n-1
\tag{22}
$$

*where $\eta_L$ and $V_L$ are the camera vector and velocity vector of the pilot, respectively. If $z_{1d}(t) = \eta_L$ and $z_{2d}(t) = V_L$, then the two input deviations can be expressed as:*

$$
\begin{cases}
z_{i1e}(t) = z_{i1}(t) - z_{1d}(t) - d_i \\
z_{i2e}(t) = z_{i2}(t) - z_{2d}(t) - dv_i
\end{cases}
\tag{23}
$$

Defining the sliding surface:

$$s_i = k_1 z_{i1e}(t) + k_2 z_{i2e}(t) \tag{24}$$

where $\dot{z}_{i1e}(t) = z_{i2e}(t)$, $\dot{z}_{i2e}(t) = U_i(t)$, take $t'$s first derivative of Equation (24), we can obtain:

$$\dot{s}_i = k_1 z_{i2e}(t) + k_2 U_i(t) \tag{25}$$

where $k_1$, $k_2 > 0$, taking the index near rate:

$$\dot{s}_i = -k s_i - \varepsilon \mathrm{sgn}(s_i) \tag{26}$$

where $k$ is the constant velocity approach coefficient and $\varepsilon$ is the exponential approach coefficient. $\varepsilon = k_3|s_i|$, $0 \le k_3 < 1$ and $\varepsilon > 0$. Bringing (26) into (25), the control quantity $U_i$ can be obtained:

$$U_i(t) = (-k k_1 z_{i1e}(t) - (k k_2 + k_1) z_{i2e}(t) - k_3|s_i|\mathrm{sgn}(s_i))/k_2 \tag{27}$$

In the model, U is the controller input of follower $i$.

### 4.2. The Design of the Formation Reconfiguration Controller

The leader–follower formation has higher formation controlling accuracy, and the communication network of formation is simple, but once the leader fails, the formation is difficult to maintain. In this section, the Hungarian algorithm in Section 2 is used to realize the autonomous formation reconfiguration control of the followers in the failed formation.

First, the state of the multi-AUVs is initialized, and the desired formation under the control of the formation controller designed in Section 4.1 is formed. The effective status of the leader in the formation is detected in real time. Once the leader is detected to be invalid, the host computer of the leader will collect and save the pose and speed information of the other effective ones. The cost coefficient matrix of the formation reconfiguration is generated in real time according to the position information. Then the Hungarian algorithm optimally assigns the generated cost coefficient matrix to minimize the cost of formation reconfiguration. Finally, according to the optimal assignment plan, the designed formation controller is used to control the formation reconfiguration.

The controller of follower $i$ in the formation is as follows:

$$U_i(t) = (-k k_1 z_{i1e}(t) - (k k_2 + k_1) z_{i2e}(t) - k_3|s_i(t)|\mathrm{sgn}(s_i(t)))/k_2 \tag{28}$$

Among them: $z_{i1e}(t)$ and $z_{i2e}(t)$ are the posture information deviations between the follower in the failed formation and the new effective leader.

### 4.3. Design of the Formation Controller under an Event-Triggered Mechanism

After formation reconfiguration, due to the increased number of followers, which increases the operation burden of some pilots, the event-triggered mechanism is used to reduce the communication frequency between follower and pilot, so as to reduce the communication pressure of the pilot AUV.

In Section 4.2, the formation controller under the continuous sequence of the multi-AUV is designed and the formation controller with the event-triggered mechanism of a multi-AUV formation system needs to be made on the basis of Equation (28).

According to the event-triggered principle, the trigger function $f_i(t)$ $i \in 1, 2, \ldots, m$ is designed for each follower AUV (assuming that there are $m$ followers in the formation), and the event-triggered sequence $t_k^i (k = 0, 1, \ldots)$ is formulated for each follower according to $f_i(t) = 0$, in which $t_k^i (k = 0, 1, \ldots)$ represents the $k$ triggering moment of the follower $i$. $\hat{z}_{1i}(t) = z_{1i}(t_k^i)$ and $\hat{z}_{2i}(t) = z_{2i}(t_k^i)$, $t \in [t_k^i, t_{k+1}^i)$ represent the sampling values of the position and velocity of follower $i$ in the time period of $[t_k^i, t_{k+1}^i)$, respectively. Combined with the formation controller (28) of the continuous time, the following can be obtained:

$$
\begin{aligned}
U_i(t) &= (-kk_1\hat{z}_{i1e}(t) - (kk_2 + k_1)\hat{z}_{i2e}(t) - k_3|k_1\hat{z}_{i1e}(t) + k_2\hat{z}_{i2e}(t)|\text{sgn}(k_1\hat{z}_{i1e}(t) + k_2\hat{z}_{i2e}(t)))/k_2 \\
&= (-kk_1(z_{1i}(t_k^i) - z_{1d}(t_k^i) + d_i) - (kk_2 + k_1)(z_{2i}(t_k^i) - z_{2d}(t_k^i)) - k_3|k_1(z_{1i}(t_k^i) - z_{1d}(t_k^i) + d_i) \\
&\quad +k_2(z_{2i}(t_k^i) - z_{2d}(t_k^i))|\text{sgn}(k_1(z_{1i}(t_k^i) - z_{1d}(t_k^i) + d_i) + k_2(z_{2i}(t_k^i) - z_{2d}(t_k^i))))/k_2, t \in [t_k^i, t_{k+1}^i)
\end{aligned}
\tag{29}
$$

The above Equation (29) is the formation controller under the event-triggered mechanism. From the formula, it can be seen that the follower can only sample and communicate the leader's pose and speed information when the event is triggered. Updating the controller's input, and the follower's controller input remains unchanged at other times.

*4.4. Analysis of Formation Stability*

For the formation controller of a multi-AUV formation system under the event-triggered mechanism, the stable convergence of an AUV under the action of the formation controller (27) should by analyzed first, and then the stability convergence problem under the action of the formation controller (29) should be analyzed.

4.4.1. Analysis of Formation Controller Stability

From the Definition 2: $\eta_i(t) \in R^5$, $V_i(t) \in R^5$, $\varepsilon_i(t) \in R^{10}$, the system status of the formation is: $\varepsilon(t) = [\xi(t); \zeta(t)] \in R^{10n}$.

In it: $\xi(t) = [\eta_1(t); \eta_2(t); \ldots; \eta_n(t)]$, $\zeta(t) = [\zeta_1(t); \zeta_2(t); \ldots; \zeta_n(t)]$.

The system state equation of a formation with n members is defined as:

$$
\begin{aligned}
\dot{\xi}(t) &= \zeta(t) \\
\dot{\zeta}(t) &= -(L_p \otimes I_5)kk_1/k_2\xi(t) - (L_V \otimes I_5)(k + k_1/k_2)\zeta(t)
\end{aligned}
\tag{30}
$$

There must be a strongly connected Laplacian matrix in the leader–follower formation system. $L_p$ and $L_V$ represent the communication contact matrix of the pose and speed in formation. $L_p$ and $L_V$ are equal, and $\otimes$ is the Kronick product.

If the error state vector is $\Delta\eta_i(t) = \eta_i(t) - \eta_L(t)$ and $\Delta V_i(t) = V_i(t) - V_L(t)$, the system error state vector of the AUV formation can be obtained as $\varsigma(t) = [\varsigma_p(t); \varsigma_V(t)]$, in which $\varsigma_p(t)$ and $\varsigma_v(t)$ are as follows:

$$
\begin{aligned}
\varsigma_p(t) &= [\Delta\eta_1(t); \Delta\eta_2(t); \cdots; \Delta\eta_n(t)] = \xi(t) - I_n \otimes \xi_L(t) \\
\varsigma_V(t) &= [\Delta V_1(t); \Delta V_2(t); \cdots; \Delta V_n(t)] = \zeta(t) - I_n \otimes \zeta_L(t)
\end{aligned}
\tag{31}
$$

Inserting Formula (31) into (30), the error state equation is:

$$
\begin{aligned}
\dot{\varsigma}_p(t) &= \varsigma_V(t) \\
\dot{\varsigma}_V(t) &= -(L_p \otimes I_5)kk_1/k_2\varsigma_p(t) - (L_V \otimes I_5)(k + k_1/k_2)\varsigma_V(t)
\end{aligned}
\tag{32}
$$

There are two nonsingular matrices which are $U_p$ and $U_v$, therefore:

$$
U_p^{-1}L_pU_p = \begin{bmatrix} 0 & \alpha_p^T \\ 0_{n-1} & H_p \end{bmatrix} = \Lambda_p,
$$

$$
U_v^{-1}L_vU_v = \begin{bmatrix} 0 & \alpha_v^T \\ 0_{n-1} & H_v \end{bmatrix} = \Lambda_v
$$

In the formula: $H_p, H_v \in R^{(n-1)(n-1)}$, $\alpha_p, \alpha_v \in R^{n-1}$.

Set $\widetilde{S}_p(t) = (U_p^{-1} \otimes I_5)S_p(t)$, $\widetilde{S}_v(t) = (U_v^{-1} \otimes I_5)S_v(t)$, and $K_1 = kk_1/k_2$, $K_2 = k + k_1/k_2$, then (32) can be transformed into:

$$
\begin{aligned}
\dot{\widetilde{\varsigma}}_p(t) &= \widetilde{\varsigma}_v(t) \\
\dot{\widetilde{\varsigma}}_v(t) &= -K_1(H_p \otimes I_5)\widetilde{\varsigma}_p(t) - K_2(H_v \otimes I_5)\widetilde{\varsigma}_v(t)
\end{aligned}
\tag{33}
$$

Since $\Lambda_p$ and $\Lambda_v$ must contain "0" terms, the system state equation can be decomposed into two terms which either contains "0" or does not. The formula (33) can be decomposed into:

$$\begin{aligned} \dot{\tilde{\varsigma}}_p^1(t) &= \tilde{\varsigma}_v^1(t) \\ \dot{\tilde{\varsigma}}_v^1(t) &= -K_1(H_p \otimes I_5)\tilde{\varsigma}_p^1(t) - K_2(H_v \otimes I_5)\tilde{\varsigma}_v^1(t) \end{aligned} \tag{34}$$

and

$$\begin{aligned} \dot{\tilde{\varsigma}}_p^2(t) &= \tilde{\varsigma}_v^2(t) \\ \dot{\tilde{\varsigma}}_v^2(t) &= -K_1(\alpha_p \otimes I_5)\tilde{\varsigma}_p^2(t) - K_2(\alpha_v \otimes I_5)\tilde{\varsigma}_v^2(t) \end{aligned} \tag{35}$$

Equations (34) and (35) are a non-zero module and zero module, respectively. First, the state error vectors $\tilde{\varsigma}^1(t) = [\tilde{\varsigma}_P^1(t)\ \tilde{\varsigma}_v^1(t)]$ and $\tilde{\varsigma}^2(t) = [\tilde{\varsigma}_P^2(t)\ \tilde{\varsigma}_v^2(t)]$ of the non-zero module and zero module are defined. If $t \to \infty$, $\tilde{\varsigma}^1(t) = 0, \tilde{\varsigma}^2(t) = 0$, the leader-follower formation is stable.

**Lemma 1.** *Supposing the matrix $S \in R^{n \times n}$ is the symmetric matrix, and $S_{11}, S_{12}, S_{21}$ and $S_{22}$ are partitioned matrices of matrix S3, the form is as follows:*

$$S = \begin{bmatrix} S_{11} & S_{12} \\ S_{21} & S_{22} \end{bmatrix}$$

If $S_{11} \in R^{r \times r}$, $S_{12} \in R^{r \times (n-r)}$, $S_{21} \in R^{(n-r) \times r}$, $S_{22} \in R^{(n-r) \times (n-r)}$, then $S < 0$ if and only if $S_{11} < 0$, $S_{11}S_{22} - S_{12}S_{21} < 0$.

- Analysis of the non-zero module stability:

  Formula (34) is transformed as:

$$\dot{\tilde{\varsigma}}^1 = A\tilde{\varsigma}^1 + B\tilde{\varsigma}^1 \tag{36}$$

where:

$$A = \begin{bmatrix} 0 & 0 \\ I & 0 \end{bmatrix} \otimes I_5, B = \begin{bmatrix} -K_2H_v & -K_1H_p \\ 0 & 0 \end{bmatrix} \otimes I_5 \tag{37}$$

Since $H_p$ and $H_v$ are positive definite matrices, there must be positive definite matrices satisfaction of $P_p$ and $P_v$:

$$\begin{aligned} P_PH_p + H_p^TP_P &= I \\ P_vH_v + H_v^TP_v &= I \end{aligned} \tag{38}$$

To prove the stability of the non-zero submodule, the Lyapunov–Razumikhin equation is defined as follows:

$$V(\tilde{\varsigma}^1(t)) = \tilde{\varsigma}^1(t)^T P \tilde{\varsigma}^1(t) \tag{39}$$

In the formula: $P$ is the positive definite angle symmetric matrix expressed as:

$$P = \begin{bmatrix} K_2P_v & -K_1P_p \\ -K_1P_p & K_2P_v \end{bmatrix} \otimes I_5 \tag{40}$$

Take the derivative of (39):

$$\dot{V}(\tilde{\varsigma}^1(t)) = 2\tilde{\varsigma}^1(t)^T P \dot{\tilde{\varsigma}}^1(t) = \tilde{\varsigma}^1(t)^T [P(A+B) + (A+B)^T P]\tilde{\varsigma}^1(t) \tag{41}$$

Defining the matrix $Q = P(A+B) + (A+B)^T P$, it can be known from Equations (37) and (40) that $Q$ must be the negative definite matrix. In conclusion:

$$\dot{V}(\tilde{\varsigma}^1(t)) = \tilde{\varsigma}^1(t)^T [P(A+B) + (A+B)^T P]\tilde{\varsigma}^1(t) = \tilde{\varsigma}^1(t)^T Q\tilde{\varsigma}^1(t) < 0 \tag{42}$$

It can be seen from Equation (39) that the designed Lyapunov—Razumikhin equation is greater than 0 and the Equation (41) shows that its derivative is less than 0. Therefore, it can be proved that the non-zero subsystem is asymptotically stable.

Similarly, the zero module is also convergent and stable. According to the stability analysis of the zero module and non-zero module, if the state error of formation is $\varsigma_p \to 0, \varsigma_v \to 0$, then the AUV formation system is stable.

4.4.2. Analysis of Formation Stability under and Event-Triggered Mechanism

Since Equation (29) consists of $\hat{s}_i(t)$, it is not conducive to analyze the formation stability, therefore Equation (29) needs to be processed as:

$$
\begin{aligned}
U_i(t) &= (-kk_1\hat{z}_{i1e}(t) - (kk_2 + k_1)\hat{z}_{i2e}(t) - k_3|\hat{s}_i(t)|\mathrm{sgn}(\hat{s}_i(t)))/k_2 \\
&\leq (-kk_1\hat{z}_{i1e}(t) - (kk_2 + k_1)\hat{z}_{i2e}(t) - k_3(k_1\hat{z}_{i1e}(t) + k_2\hat{z}_{i2e}(t))))/k_2 \\
&= -((kk_1 + k_3k_1)\hat{z}_{i1e}(t) + (kk_2 + k_3k_2)\hat{z}_{i2e}(t))/k_2
\end{aligned}
\tag{43}
$$

If: $e_{z_{1i}}(t) = \hat{z}_{1i}(t) - z_{1i}(t)$, $e_{z_{2i}}(t) = \hat{z}_{2i}(t) - z_{2i}(t)$, $e_{z_{1d}}(t) = \hat{z}_{1d}(t) - z_{1d}(t) + d$, $e_{z_{2d}}(t) = \hat{z}_{2d}(t) - z_{2d}(t)$

Bring the pose error and velocity difference of follower $i$ in into Equation (43) and obtain:

$$
\begin{aligned}
U_i(t) &= -K_1(z_{1i}(t) - z_{1d}(t) + e_{z_{1i}}(t) - e_{z_{1d}}(t)) - K_2(z_{2i}(t) - z_{2d}(t) \\
&\quad + e_{z_{2i}}(t) - e_{z_{2d}}(t)), t \in [t_k^i, t_{k+1}^i)
\end{aligned}
\tag{44}
$$

In the formula, $K_1 = (kk_1 + k_3k_1)/k_2$, $K_2 = (kk_2 + k_3k_2)/k_2$.

Combining Equation (8), we can obtain:

$$
\begin{cases}
\dot{z}_{1i}(t) = z_{2i}(t) \\
\dot{z}_{2i}(t) = -K_1(z_{1i}(t) - z_{1d}(t) + e_{z_{1i}}(t) - e_{z_{1d}}(t)) - K_2(z_{2i}(t) - z_{2d}(t) \\
\qquad\quad + e_{z_{2i}}(t) - e_{z_{2d}}(t)), t \in [t_k^i, t_{k+1}^i)
\end{cases}
\tag{45}
$$

Therefore, the entire formation system of followers can be described as:

$$
\begin{cases}
\dot{z}_1(t) = z_2(t) \\
\dot{z}_2(t) = -L(K_1z_1(t) + K_2z_2(t) + K_1e_{z_1}(t) + K_2e_{z_2}(t))
\end{cases}
\tag{46}
$$

where: $L$ is the communication topology between the follower and formation, and:

$$
\begin{aligned}
z_1(t) &= (z_{11}(t)^T, z_{12}(t)^T, \ldots, z_{1m}(t)^T)^T \\
z_2(t) &= (z_{21}(t)^T, z_{22}(t)^T, \ldots, z_{2m}(t)^T)^T \\
e_{z_1}(t) &= (e_{z_{11}}(t)^T, e_{z_{12}}(t)^T, \ldots, e_{z_{1m}}(t)^T)^T \\
e_{z_2}(t) &= (e_{z_{21}}(t)^T, e_{z_{22}}(t)^T, \ldots, e_{z_{2m}}(t)^T)^T
\end{aligned}
$$

The position and velocity states of all the followers in the formation system relative to their pilots are decomposed:

$$
\begin{aligned}
z_1(t) &= z_{1d}(t) + \varepsilon_{z1}(t) - d(t) \\
z_2(t) &= z_{2d}(t) + \varepsilon_{z2}(t) - dv(t)
\end{aligned}
\tag{47}
$$

$z_{1d}(t)$, $z_{2d}(t)$, $d(t)$ and $dv(t)$ are the state values of the leader's position, velocity, and formation constraints in the fixed coordinate system at time $t$, respectively. $\varepsilon_{z1}(t)$ and $\varepsilon_{z2}(t)$ are the state components of non-formation.

According to Equation (47), the following matrix can be obtained:

$$
\begin{bmatrix} \dot{\varepsilon}_{z1}(t) \\ \dot{\varepsilon}_{z2}(t) \end{bmatrix} = \begin{bmatrix} 0_{m \times m} & I_m \\ -K_1 L & -K_2 L \end{bmatrix} \begin{bmatrix} z_{1d}(t) + z_{2d}(t) + \varepsilon_{z1}(t) \\ z_{2d}(t) + \varepsilon_{z2}(t) \end{bmatrix} - \begin{bmatrix} 0_{m \times m} & 0_{m \times m} \\ K_1 L & K_2 L \end{bmatrix} e(t) - \begin{bmatrix} \varepsilon_{z2}(t) \\ 0_{m \times m} \end{bmatrix} \tag{48}
$$

In the formula, $e(t) = (e_{z1}(t)^T, e_{z2}(t)^T)^T \in R^{2 \times m}$, so from the above formula, we can obtain:

$$
\begin{cases} \dot{\varepsilon}_{z1}(t) = \varepsilon_{z2}(t) \\ \dot{\varepsilon}_{z2}(t) = -L(K_1 \varepsilon_{z1}(t) + K_2 \varepsilon_{z2}(t) + K_1 e_{z1}(t) + K_2 e_{z2}(t)) \end{cases} \tag{49}
$$

From the literature [13], the study on the stability of the multi-AUV formation system (8) is equivalent to the stability of system (49). Therefore, as long as the system (49) is proved to be stable, the system (8) can be proved to be stable by Equation (47) and the Lyapunov stability theory can be used to prove that the closed-loop system (8) is convergent.

For simplicity, $\{t_0^i, t_1^i, \ldots, t_k^i\}$ represents a series of follower $i$'s trigger moments during the formation process, and the iteration update of the trigger moment $t_k^i$ is as follows:

$$
t_{k+1}^i = \inf \left\{ t > t_k^i : f_i(t) \geq 0 \right\} \tag{50}
$$

The event-triggered function is designed as follows:

$$
f_i(t) = \left| e_{z1i}(t) \right|_2 + \left| e_{z2i}(t) \right|_2 + \left| e_{z1d}(t) \right|_2 + \left| e_{z2d}(t) \right|_2 - \frac{\sigma_i \alpha}{\rho} (\lambda_2(L) - 2\alpha \rho m)(\left| \varepsilon_{z1i}(t) \right|_2 + \left| \varepsilon_{z2i}(t) \right|_2) \tag{51}
$$

**Theorem 1.** *If the multi-AUV formation system (8) is under the action of a formation controller (43) which adopts the leader–follower formation strategy, then the follower i can meet the following condition of an event-triggered mechanism (51):*

$$
\lambda_2(L) - 2\alpha \rho m > 0 \tag{52}
$$

*where $\rho = \max\{K_1, K_2\}$, $0 < \sigma_i < 1$, and $\alpha > 0$ is constant, so for all follower AUVs:*

$$
\lim_{t \to \infty} z_{1i}(t) = z_{1d}(t) - d_i(t)
$$
$$
\lim_{t \to \infty} z_{2i}(t) = z_{2d}(t) - dv_i(t), i = 1, 2, \ldots, m
$$

*This means the formation system (8) can converge stably.*

**Proof 1.** For the closed-loop system (49), the following Lyapunov equation is constructed:

$$
V(t) = \frac{1}{2} \varepsilon^T(t) P \varepsilon(t) \tag{53}
$$

In the equation, $\varepsilon(t) = (\varepsilon_{z1}(t)^T, \varepsilon_{z2}(t)^T)^T \in R^{2 \times m}$, $P = \begin{pmatrix} (K_1 + K_2)L & I_m \\ -I_m & I_m \end{pmatrix}$, $m$ represents the number of followers in the formation.

Since $\varepsilon_{z1}(t)^T L \varepsilon_{z1}(t) \geq \lambda_2(L) \left\| \varepsilon_{z1}(t) \right\|^2$, therefore:

$$
V(t) = \frac{K_1 + K_2}{2} \varepsilon^T(t) L \varepsilon(t) + \frac{1}{2} \varepsilon_{z2}^T(t) P \varepsilon_{z2}(t) \geq 0 \tag{54}
$$

From the derivation of Equation (53) along the trajectory of system (49), we can obtain:

$$
\begin{aligned}
\dot{V}(t) &= \varepsilon^T(t)P\varepsilon(t) \\
&= -\varepsilon^T(t)\begin{bmatrix} K_1 L & -K_1 L \\ K_1 L & K_2 L + I_m \end{bmatrix}\varepsilon(t) - \varepsilon^T\begin{bmatrix} K_1 L & K_2 L \\ K_1 L & K_2 L \end{bmatrix}e(t) \\
&= -\varepsilon_{z1}^T(t)L\varepsilon_{z1}(t) - \varepsilon_{z2}^T(t)(K_2 L + I_m)\varepsilon_{z2}(t) - (\varepsilon_{z1}^T(t) + \varepsilon_{z2}^T(t))L(K_1 e_{z1}(t) + K_2 e_{z2}(t)) \\
&\leq -\lambda_2(L)||\varepsilon_{z1}(t)||_2 - (\lambda_2(L) + 1)|\varepsilon_{z2}(t)||_2 - (\varepsilon_{z1}^T(t) + \varepsilon_{z2}^T(t))L(K_1 e_{z1}(t) + K_2 e_{z2}(t)) \\
&= -\lambda_2(L)\sum_{i=1}^{m}|\varepsilon_{z1i}(t)|^2 - (\lambda_2(L) + 1)\sum_{i=1}^{m}|\varepsilon_{z2i}(t)|^2 - (\varepsilon_{z1}^T(t) + \varepsilon_{z2}^T(t))L(K_1 e_{z1}(t) + K_2 e_{z2}(t))
\end{aligned}
\tag{55}
$$

Using the inequality property $\left|xy\right| \leq \frac{\alpha}{2}x^2 + \frac{1}{2\alpha}y^2$, we can obtain:

$$
\begin{aligned}
&-(\varepsilon_{z1}^T(t) + \varepsilon_{z2}^T(t))L(K_1 e_{z1}(t) + K_2 e_{z2}(t)) \\
&= -\sum_{i=1}^{m}(\varepsilon_{z1i}^T(t) + \varepsilon_{z2i}^T(t))(K_1(e_{z1i}(t) - e_{z1d}(t)) + K_2(e_{z2i}(t) - e_{z2d}(t))) \\
&= -\sum_{i=1}^{m} m(\varepsilon_{z1i}^T(t) + \varepsilon_{z2i}^T(t))(K_1 e_{z1i}(t) + K_2 e_{z2i}(t)) + \sum_{i=1}^{m}(\varepsilon_{z1d}^T(t) + \varepsilon_{z2d}^T(t))(K_1 e_{z1d}(t) + K_2 e_{z2d}(t)) \\
&\leq 2\alpha\rho\sum_{i=1}^{m} m(|\varepsilon_{z1i}(t)|^2 + |\varepsilon_{z2i}(t)|^2) + \frac{\rho}{\alpha}\sum_{i=1}^{m}(|e_{z1i}(t)|^2 + |e_{z2i}(t)|^2 + |e_{z1d}(t)|^2 + |e_{z2d}(t)|^2)
\end{aligned}
\tag{56}
$$

In the equation: $\rho = \max\{K_1, K_2\}$. Therefore:

$$
\begin{aligned}
\dot{V}(t) &\leq -\lambda_2(L)\sum_{i=1}^{m}|\varepsilon_{z1i}(t)|^2 - (\lambda_2(L) + 1)\sum_{i=1}^{m}|\varepsilon_{z2i}(t)|^2 \\
&\quad -(\varepsilon_{z1}^T(t) + \varepsilon_{z2}^T(t))L(K_1 e_{z1}(t) + K_2 e_{z2}(t)) \\
&= -\sum_{i=1}^{m}(\lambda_2(L) - 2\alpha\rho m)|\varepsilon_{z1i}(t)|^2 - \sum_{i=1}^{m}(\lambda_2(L) + 1 - 2\alpha\rho m)|\varepsilon_{z2i}(t)|^2 + \\
&\quad \frac{\rho}{\alpha}\sum_{i=1}^{m}(|e_{z1i}(t)|^2 + |e_{z2i}(t)|^2 + |e_{z1d}(t)|^2 + |e_{z2d}(t)|^2) \\
&\leq -\sum_{i=1}^{m}(\lambda_2(L) - 2\alpha\rho m)(|\varepsilon_{z1i}(t)|^2 + |\varepsilon_{z2i}(t)|^2) \\
&\quad -\frac{\rho}{\alpha}\sum_{i=1}^{m}(|e_{z1i}(t)|^2 + |e_{z2i}(t)|^2 + |e_{z1d}(t)|^2 + |e_{z2d}(t)|^2)
\end{aligned}
\tag{57}
$$

According to the event-triggered mechanism (51), when all followers meet the event-triggered conditions:

$$
f_i(t) \leq 0
\tag{58}
$$

Namely,

$$
|e_{z1i}(t)|^2 + |e_{z2i}(t)|^2 + |e_{z1d}(t)|^2 + |e_{z2d}(t)|^2 \leq \frac{\sigma_i \alpha}{\rho}(\lambda_2(L) - 2\alpha\rho m)\left(\left|\varepsilon_{z1i}(t)\right|_2 + \left|\varepsilon_{z2i}(t)\right|_2\right)
$$

Bring this into (57) to obtain:

$$
\dot{V}(t) \leq -\sum_{i=1}^{m}(1 - \sigma_i)(|\varepsilon_{z1i}(t)|^2 + |\varepsilon_{z2i}(t)|^2) \leq 0
\tag{59}
$$

□

From Equations (54) and (59), we know that the closed-loop system (49) is asymptotically stable. $\varepsilon_{z1}(0)$ and $\varepsilon_{z2}(0)$ under any initial state will meet:

$$
\begin{cases} \lim\limits_{t\to\infty} \varepsilon_{z1}(t) = 0 \\ \lim\limits_{t\to\infty} \varepsilon_{z2}(t) = 0 \end{cases}
$$

Therefore, the formation system (8) is asymptotically convergent and stable.

According to Equation (51), the designed event-triggered function is related to the sampling error of the follower and leader, and the real-time error of the follower. Therefore,

there will be a problem. When the formation is completed, if the sampling error of the follower is equal to the real-time error and $\sigma_i \alpha(\lambda_2(L) - 2\alpha\rho m)/\rho$ is small (usually less than 1), this would cause a frequent sampling update after completing formation, which would then cause frequent information interactions between the followers and pilots. Inspired by the time-intermittent communication based on threshold in the literature [13], the designed event-triggered function (51) is improved as follows:

$$t_{k+1}^i = \inf\left\{ t > t_k^i : |e_{z1i}(t)|^2 + |e_{z2i}(t)|^2 > \omega \,\&\&\, f_i(t) \geq 0 \right\} \tag{60}$$

The Equation (60) shows that the trigger time will be updated when the sampling error square of the follower's position and velocity is greater than the threshold value $\omega$ which is a constant and generally $0 < \omega < 1$. The larger the value of $\omega$ is, the greater the controller updating interval of the follower will be. This will have an influence on the control effect if the value of $\omega$ is too large. After meeting the conditions of the sampling error, this will determine whether the sampling information of the position and speed of the follower and the leader meet the conditions of the event-triggered mechanism or not. If both conditions are satisfied, the follower will update the controller input, which can effectively solve the problem of the mechanism being frequently triggered after completing the formation.

In order to prevent the Zeno phenomenon from occurring in the system (8) during the whole process of the event-triggered control, the following theorem is proved [25].

**Theorem 2.** *The controller (29) has an effect on the system (8), therefore the interval ($t_{k+1}^i - t_k^i$) between any two consecutive event-triggered moments is not less than:*

$$\tau_q = \Gamma_q \left[ c \left( \sqrt{2m}\rho\|L\| + \sqrt{mN} \right) \left( c\sqrt{2N}\|\mathbf{D} + A\| + \Gamma_q \right) \right]^{-1} \tag{61}$$

*In the formula, $c > 1$ is a constant, and $\Gamma_q = \frac{\sigma_i \alpha}{\rho}(\lambda_2(L) - 2\alpha\rho m)^{\frac{1}{2}}$.*

**Proof 2.** We can obtain:

$$\sum \left( |e_{z_{1i}}(t)| + |e_{z_{1d}}(t)| + |e_{z_{2i}}(t)| + |e_{z_{2d}}(t)| \right)$$

This is the *i*th component of $|\;D + A \quad D + A| \cdot |e(t)\;|$ , therefore:

$$\begin{aligned}
\sum\sum \left( |e_{z_{1i}}(t)|^2 + |e_{z_{1d}}(t)|^2 + |e_{z_{2i}}(t)|^2 + |e_{z_{2d}}(t)|^2 \right) \\
\leq \| |D + A\ D + A| \cdot |e(t)| \|^2 \\
\leq 2\|\mathbf{D} + A\|^2 \|\mathbf{e}(t)\|^2
\end{aligned} \tag{62}$$

If $q = \mathrm{argmax}_{i \in \nu} \left( |\varepsilon_{z_1}|^2 + |\varepsilon_{z_2}|^2 \right)$, we can obtain:

$$\begin{aligned}
\frac{\sum \left( |e_{z_{1i}}(t)|^2 + |e_{z_{1d}}(t)|^2 + |e_{z_{2i}}(t)|^2 + |e_{z_{2d}}(t)|^2 \right)}{\left( |\varepsilon_{z_1}|^2 + |\varepsilon_{z_2}|^2 \right)} \\
\leq \frac{2N\|\mathbf{D} + A\|^2 \|\mathbf{e}(t)\|^2}{\|\varepsilon(t)\|^2}
\end{aligned} \tag{63}$$

while the derivative of $\frac{\|e(t)\|}{\|\varepsilon(t)\|}$

$$
\begin{aligned}
\frac{\mathrm{d}}{\mathrm{d}t}\frac{\|e(t)\|}{\|\varepsilon(t)\|} &= \frac{\mathrm{d}}{\mathrm{d}t}\frac{\left(e^T(t)e(t)\right)^{1/2}}{\left(\varepsilon^T(t)\varepsilon(t)\right)^{1/2}} \\
&= \frac{e^T(t)\dot{e}(t)}{\|e(t)\|\|\varepsilon(t)\|} - \frac{\varepsilon(t)^r\dot{\varepsilon}(t)\|e(t)\|}{\|\varepsilon(t)\|^2\|\varepsilon(t)\|} \\
&\leq \frac{\|e(t)\|}{\|\varepsilon(t)\|} + \frac{\|\dot{\varepsilon}(t)\|\|e(t)\|}{\|\varepsilon(t)\|^2}
\end{aligned}
\tag{64}
$$

Since:

$$
\|\dot{e}(t)\| = \left\| \begin{array}{c} \dot{\varepsilon}_{z_1}(t) + b1 \\ \dot{\varepsilon}_{z_2}(t) \end{array} \right\| \leq c\|\dot{\varepsilon}(t)\|
$$

In the formula: $c > 1$ is a constant.
Therefore:

$$
\begin{aligned}
\frac{\mathrm{d}}{\mathrm{d}t}\frac{\|e(t)\|}{\|\varepsilon(t)\|} &\leq \frac{c\|\dot{\varepsilon}(t)\|}{\|\varepsilon(t)\|} + \frac{\|\dot{\varepsilon}(t)\|\|e(t)\|}{\|\varepsilon(t)\|^2} \\
&= \frac{1}{\|\varepsilon(t)\|}\left(c + \frac{\|e(t)\|}{\|\varepsilon(t)\|}\right)\varepsilon(t)\left\|\left[\begin{array}{cc} \mathbf{0}_{N\times N} & \mathbf{I}_N \\ \mathbf{0}_{N\times N} & \mathbf{0}_{N\times N} \end{array}\right] - \left[\begin{array}{cc} \mathbf{0}_{N\times N} & \mathbf{0}_{N\times N} \\ k_1\mathbf{L} & k_2\mathbf{L} \end{array}\right](\varepsilon(t) + \mathbf{e}(t))\right\| \\
&\leq \left(c + \frac{\|e(t)\|}{\|\varepsilon(t)\|}\right) \cdot \frac{\sqrt{mN}\|\varepsilon(t)\| + \sqrt{2m\rho}\|L\|(\|\varepsilon(t)\| + \|e(t)\|)}{\|\varepsilon(t)\|} \\
&= \sqrt{2m\rho}\|L\|\left(1 + \frac{\|e(t)\|}{\|\varepsilon(t)\|}\right)^2 + \sqrt{mN}\left(c + \frac{\|e(t)\|}{\|\varepsilon(t)\|}\right) \\
&\leq \left(\sqrt{2m\rho}\|L\| + \sqrt{mN}\right)\left(c + \frac{\|e(t)\|}{\|\varepsilon(t)\|}\right)^2
\end{aligned}
\tag{65}
$$

Then, we can obtain:

$$
\sqrt{2N}\|D + A\|\frac{c^2(\sqrt{2m\rho}\|L\| + \sqrt{mN})\tau_q}{1 - c(\sqrt{2m\rho}\|L\| + \sqrt{mN})\tau_q} = \left(\frac{\sigma_i\alpha}{\rho}(\lambda_2(L) - 2\alpha\rho m)\right)^{\frac{1}{2}}
\tag{66}
$$

The solution is:

$$
\tau_q = \Gamma_q\left[c\left(\sqrt{2m\rho}\|L\| + \sqrt{mN}\right)\left(c\sqrt{2N}\|\mathbf{D} + A\| + \Gamma_q\right)\right]^{-1}
$$

In the formula: $\Gamma_q = \frac{\sigma_i\alpha}{\rho}(\lambda_2(L) - 2\alpha\rho m)^{\frac{1}{2}}$. The Proof is completed. $\square$

## 5. Simulation Verification and Analysis

In order to verify the effectiveness of the above designed algorithm, a simulation verification was carried out in three cases according to different numbers of leaders and followers. The trigger time and rate of the follower AUVs' controller under the event-triggered formation controller after the formation reconfiguration were finally verified.

### 5.1. When the Number of Followers Is Equal to the Number of Effective Leaders

If $0 \leq t \leq 600$, the trajectories of five leaders are shown as Table 1.

**Table 1.** Trajectory of the leader.

|  | Leader 0 | Leader 01 | Leader 02 | Leader 03 |
|---|---|---|---|---|
| $x_p$ | 0 | 30 | −30 | 60 |
| $y_p$ | 0.6 t | 0.6 t | 0.6 t | 0.6 t |
| $z_p$ | 0 | 0 | 0 | 0 |

If leader 0 fails at a time of $t = 300$ s and the other leaders still work normally, then four followers of leader 0's formation are randomly deployed. If the initial range of $y_i(0)$ and $x_i(0)$ is [0, 10] m, and the initial range of $z_i(0)$ is [−5, 0] m, then the initial range of

the initial attitude $\theta_i(0)$ is [0, 1] rad, and the initial range of the heading angle $\psi_i(0)$ is [0, $\pi$]. The initial range of the longitudinal velocity $u_i(0)$ is [0, 0.5] m/s, and all other speeds are initialized to 0 m/s. The controller parameters are $k = 1, k_1 = 0.2, k_2 = 0.8, k_3 = 0.007$, $R = 10$ m, and $\beta = \pi/4$ in formation.

The additional cost matrix $C_e$ is as follows:

$$C_e = \begin{bmatrix} 5 & 3 & 2 & 0 \\ 10 & 0 & 4 & 3 \\ 1 & 5 & 0 & 2 \\ 0 & 7 & 3 & 2 \end{bmatrix}$$

The simulation results are shown in Table 2.

**Table 2.** Formation reconfiguration cost table.

| Leader | UUV 1 | UUV 2 | UUV 3 | UUV 4 |
|--------|-------|-------|-------|-------|
| Leader 01 | 23.0909 | 20.6316 | 28.3197 | 28.9923 |
| Leader 02 | 27.9010 | 22.9578 | 23.5096 | 23.6661 |
| Leader 03 | 6.5087 | 6.4433 | 11.2257 | 18.9076 |
| Leader 04 | 10.8070 | 12.8731 | 6.9274 | 5.5913 |

Table 2 is the cost table of redistributing the followers in the failed formation to the formation of the effective Leader 01, Leader 02, Leader 03, and Leader 04 at the time of $t = 300$. The redistribution scheme of the formation leader can be observed in the table: follower No. 1 joins the formation of leader No. 3, and follower No. 2 joins the formation of leader No. 1. Followers No. 3 and No. 4 join the formation of leader No. 2, and No. 4, respectively. The total amount of cost for the formation reconfiguration is 56.2412.

Figure 5 shows the simulation of the respective degrees of formation.

Figure 5a is the trajectory diagram of the multi-AUV formation, and Figure 5b is the projection diagram of the multi-AUV formation trajectory in the horizontal plane. It shows that the follower AUV can complete and maintain the desired formation from the random initial position through the influence of the formation controller. It can be seen from Figure 5c,d that when $t = 300$ and the speed of the failed leader 0 is 0, the followers in the formation start to restructure the formation. During the reconfiguration process, the vertical velocity of the follower will oscillate, but the amplitude of the oscillation is very small and even almost zero. The follower's other speed state quantities fluctuate slightly, and the speed error converges to near zero.

*5.2. When the Number of Followers Is Less than the Number of Effective Leaders*

The parameters of the simulation verification environment and formation controller are the same as those set out in the previous section. If the number of followers in the formation of leader 0 is three, the additional matrix is changed into:

$$C_e = \begin{bmatrix} 5 & 3 & 2 \\ 10 & 0 & 3 \\ 2 & 4 & 0 \\ 0 & 3 & 5 \end{bmatrix}$$

The simulation results are shown in Table 3.

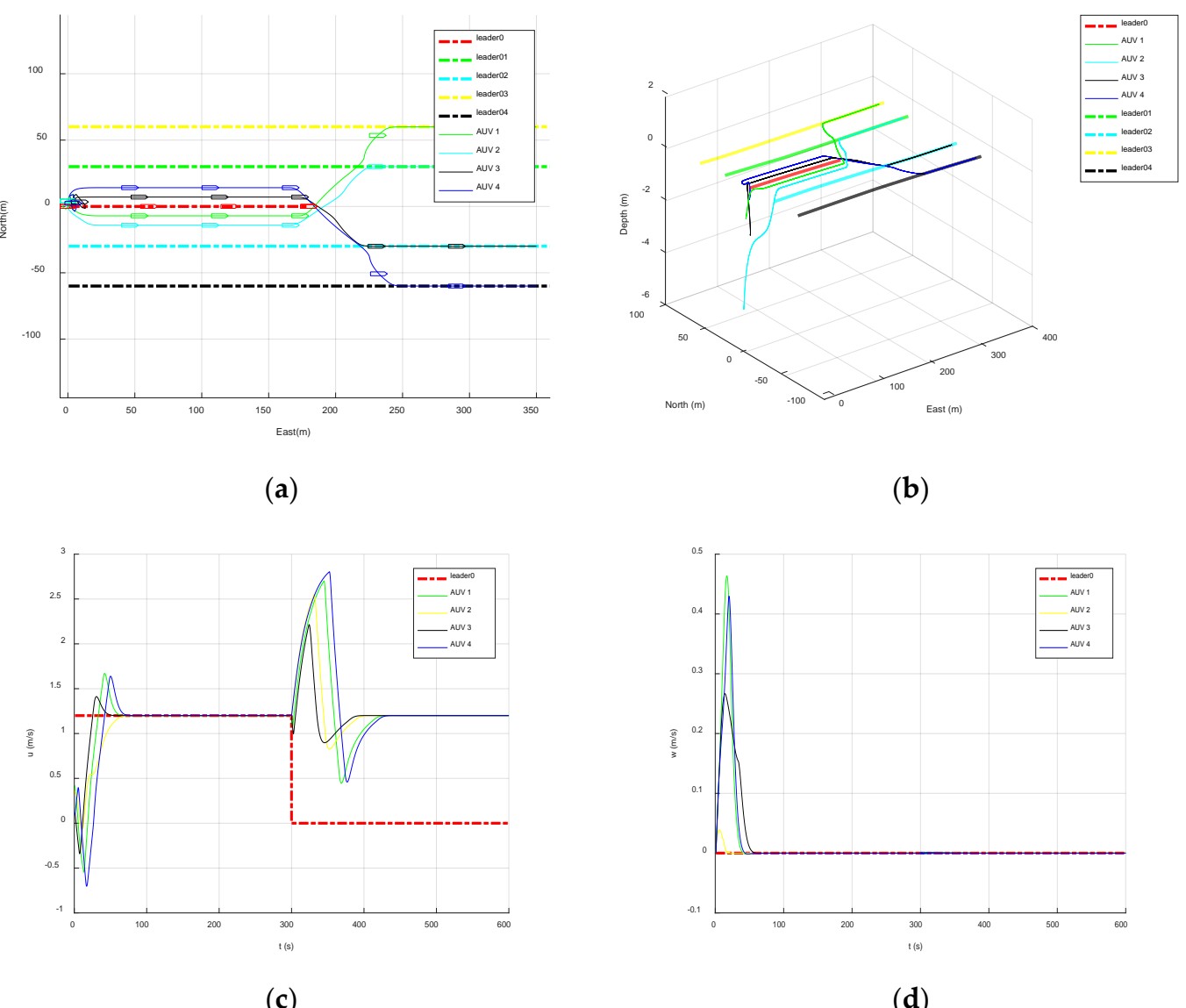

**Figure 5.** AUV formation reconfiguration. (**a**) Horizontal projection map of formation reconfiguration. (**b**) Three-dimensional trajectory map of formation reconfiguration. (**c**) AUV longitudinal speed. (**d**) AUV vertical speed.

**Table 3.** Formation reconfiguration cost table.

| Leader | UUV 1 | UUV 2 | UUV 3 |
| --- | --- | --- | --- |
| Leader 01 | 12.0184 | 17.3403 | 26.7559 |
| Leader 02 | 10.2083 | 26.6665 | 20.9458 |
| Leader 03 | 22.6007 | 5.2655 | 10.6619 |
| Leader 04 | 14.3024 | 16.5818 | 9.3635 |

Table 3 shows the redistribution scheme of the leaders: followers No. 1, No. 2 and No. 3 join the formation of leaders No. 2, No. 3 and No. 4, respectively. The total cost of reconfiguration is 24.8373.

Figure 6 is the trajectory diagram of the multi-AUV formation. The simulation diagram shows that the assignment in which the number of followers is less than the number of effective leaders can be achieved by the improved Hungarian algorithm.

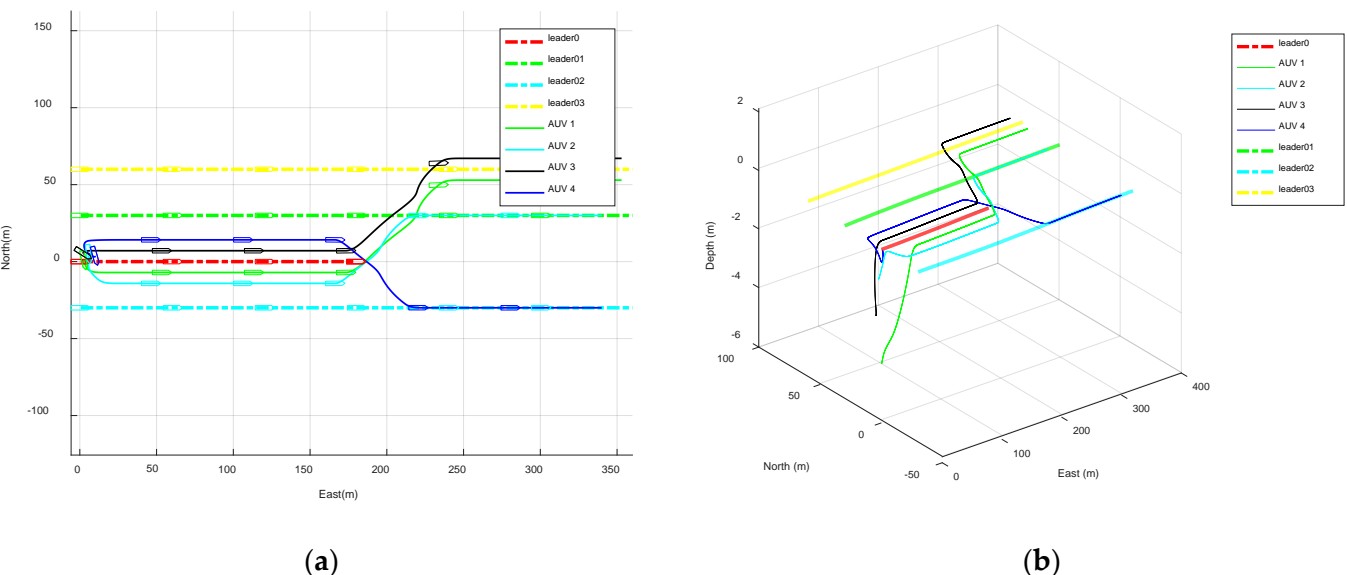

**(a)**                                                        **(b)**

**Figure 6.** AUV formation reconfiguration. (**a**) Horizontal projection map of formation reconfiguration. (**b**) Three-dimensional trajectory map of formation reconfiguration.

*5.3. When the Number of Followers Is More than the Number of Effective Leaders*

The parameters of the simulation verification environment and formation controller are the same as those set out in the previous section. If the number of effective leaders is 3, the additional matrix is changed into:

$$C_e = \begin{bmatrix} 0 & 2 & 0 & 0 \\ 3 & 0 & 0 & 0 \\ 0 & 10 & 0 & 4 \end{bmatrix}$$

The simulation results are shown in Table 4.

**Table 4.** Formation reconfiguration cost table.

| Leader | UUV 1 | UUV 2 | UUV 3 | UUV 4 |
|---|---|---|---|---|
| Leader 01 | 23.5843 | 19.6624 | 29.6931 | 29.9375 |
| Leader 02 | 28.7509 | 26.9014 | 24.5265 | 20.6985 |
| Leader 03 | 13.9954 | 18.5044 | 19.5851 | 24.6031 |

According to Table 4, the redistribution scheme of the leaders is as follows: followers No. 1, No. 2, No. 3 and No. 4 join the formation of leaders No. 3, No.1, No.3 and No. 2, respectively. The total cost of formation reconfiguration is 79.9414.

Figure 7 is the trajectory diagram of the multi-AUV formation. The simulation diagram shows that the assignment in which the number of followers is more than the number of effective leaders can be achieved by the improved Hungarian algorithm. From the simulation diagram projected on the horizontal plane, we can observe that when the number of followers needing formation reconfiguration exceeds the number of effective leaders, some leaders will receive several followers.

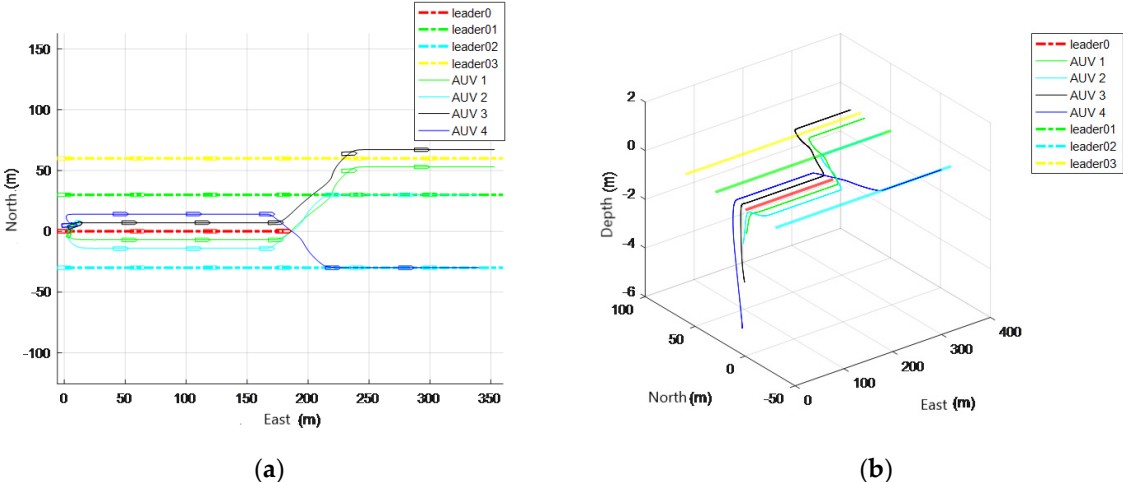

**Figure 7.** AUV formation reconfiguration. (**a**) Horizontal projection map of formation reconfiguration. (**b**) Three-dimensional trajectory map of formation reconfiguration.

### 5.4. Simulation Verification and Analysis of the Event-Triggered Mechanism

A conventional formation control experiment was used to test the performance of the designed event-triggered mechanism. The desired trajectory of the leader is:

$$
\begin{cases}
\begin{cases}
x_p = 0 \\
y_p = 0.2t \quad 0 \le t \le 300 \\
z_p = 0
\end{cases} \\
\begin{cases}
x_p = 60\cos((t-300) \times 2\pi/1000) - 60 \\
y_p = 60\sin((t-300) \times 2\pi/1000) + 60 \quad 300 < t \le 1000 \\
z_p = -0.03(t-300)
\end{cases}
\end{cases}
\tag{67}
$$

There were three followers in the formation which were deployed randomly: the initial range of $y_i(0)$ and $x_i(0)$ is [0, 5] m, and the initial range of $z_i(0)$ is [−3, 3] m. The initial range of initial attitude and heading angle $\psi_i(0)$ is [0, 1] rad. The initial range of longitudinal velocity $u_i(0)$ is [0, 0.5] m/s, and all the other speeds are initialized to 0 m/s. The controller parameters are $k = 1$, $k_1 = 0.2$, $k_2 = 0.8$, $k_3 = 0.007$, and $R = 10$m, $\beta = \pi/4$ of the formation. In the simulation environment, the flow velocity of ocean current was 0.2 m/s and the flow direction were 30°.

Event-triggered parameters: $\lambda_2(L) = 1$, $m = 3$, $K_1 = 0.25175$, $K_2 = 1.007$, $\alpha = 0.05$, $\sigma_i = 0.9$ and $\varpi = 0.2$.

The simulation results are shown in Table 5.

**Table 5.** Follower AUV event-triggered time and rate.

| Follower | Sampling Frequency | Triggering Times [13] | Triggering Rate [13] | Triggering Times in the Thesis | Triggering Rate in the Thesis |
|---|---|---|---|---|---|
| Follower 01 | | 4395 | 43.95% | 882 | 8.82% |
| Follower 02 | 10,000 | 5248 | 52.48% | 960 | 9.6% |
| Follower 03 | | 5419 | 54.19% | 1055 | 10.55% |

Figure 8 is the effect diagram of the formation trajectory tracking under the control of the formation controller based on the event-triggered mechanism, which shows that the designed formation controller could complete the desired formation task. In addition, the formation controller triggered by time can play an important role in the direct route formation and complex formation in a three-dimensional space.

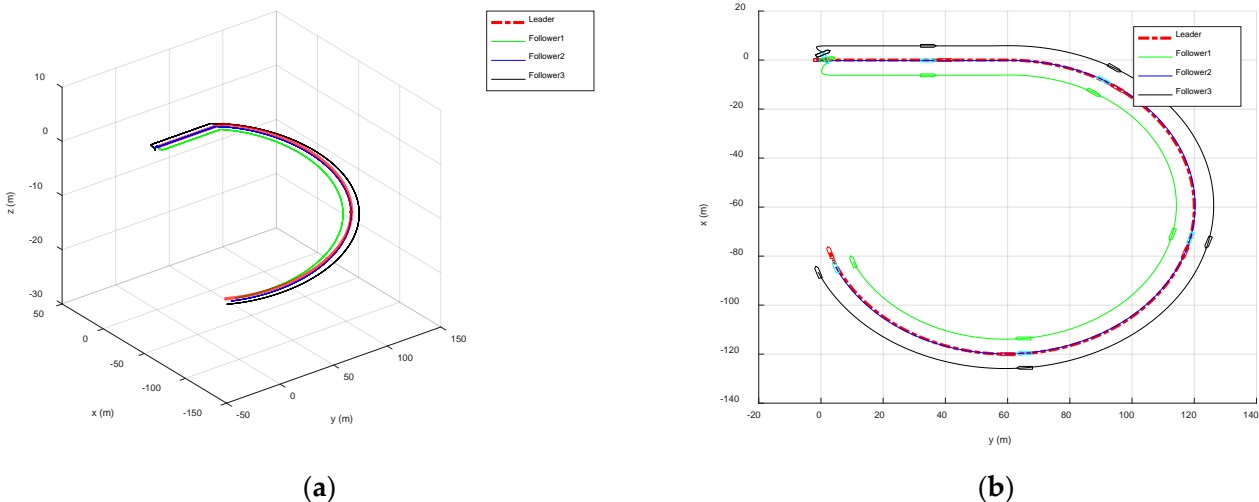

(**a**)                                        (**b**)

**Figure 8.** AUV formation trajectory. (**a**) AUV three-dimensional motion trajectory. (**b**) Horizontal projection of AUV trajectory.

Figure 9 is a simulation diagram of the pose state of each AUV in the formation, and Figure 10 is a simulation diagram of the velocity state of each AUV in the formation. From Figures 9 and 10, it can be seen that each state of each AUV changes steadily during the formation process, that each degree of convergence is fast, and that the convergence effect is good, which confirms the good control quality of the event-triggered controller designed in this paper.

Figure 11 is the triggering cases of the formation controller designed by the event-triggered function in this paper. Figure 12 is the triggering cases of the formation controller designed by the event-triggered function in this document [13].

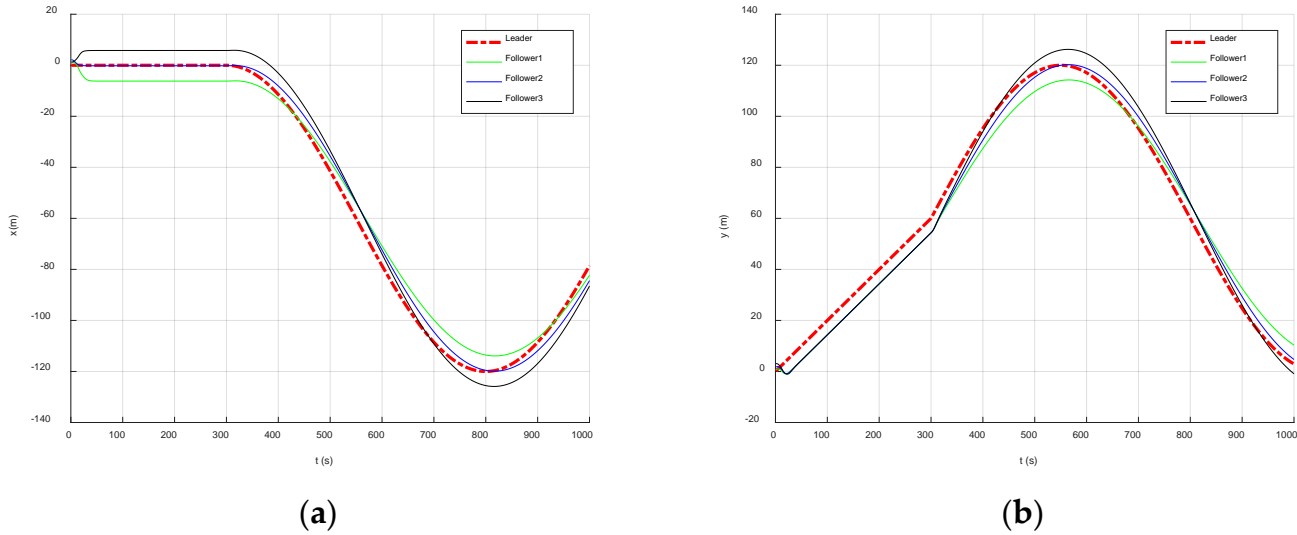

(**a**)                                        (**b**)

**Figure 9.** *Cont.*

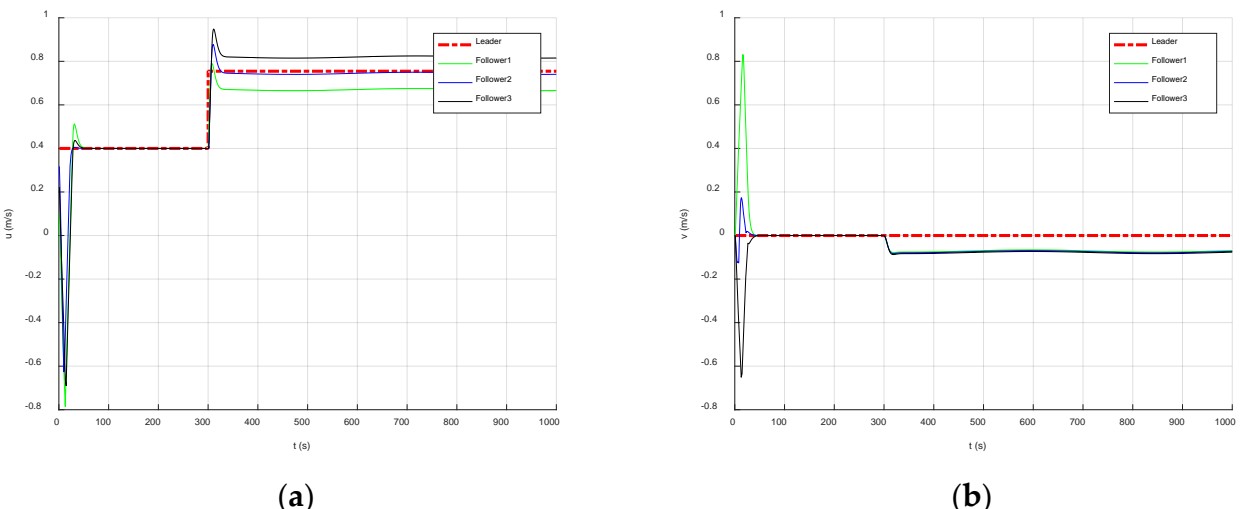

**Figure 9.** AUVs' position status. (**a**) Northbound trajectory of AUV formation. (**b**) AUV formation eastward trajectory. (**c**) AUV formation vertical trajectory. (**d**) AUVs trim angle. (**e**) AUVs heading angle.

**Figure 10.** *Cont*.

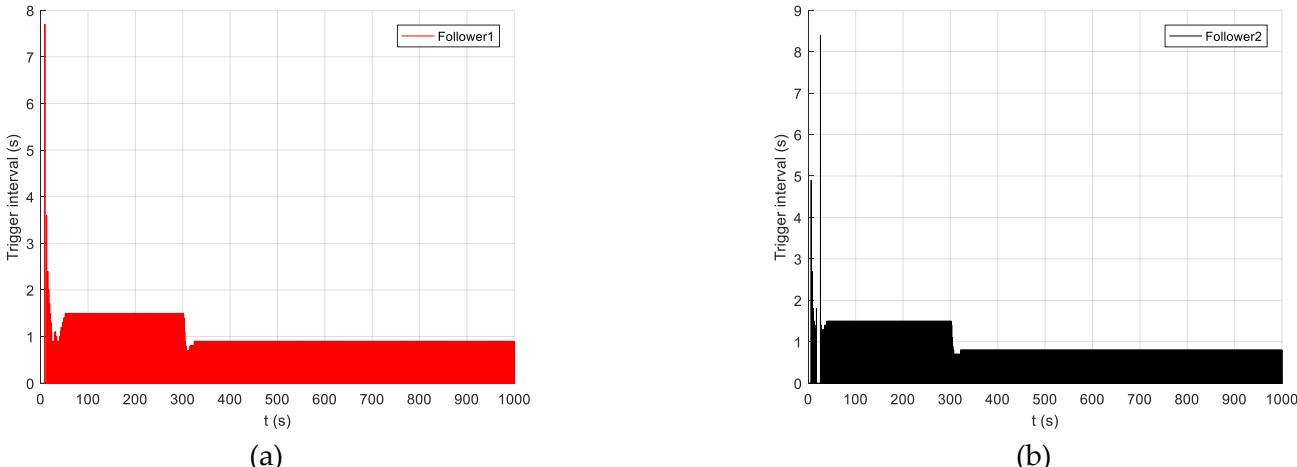

**Figure 10.** AUVs' speed status. (**a**) AUVs heading speed. (**b**) AUVs lateral speed. (**c**) AUVs vertical speed. (**d**) AUVs trim angular velocity. (**e**) AUVs heading angular velocity.

**Figure 11.** *Cont.*

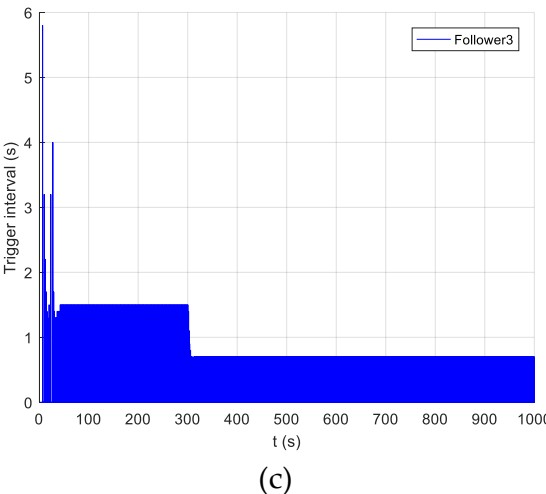

(c)

**Figure 11.** The follower event-triggered situation under the algorithm in this paper. (**a**) Follower 1 triggering, (**b**) Follower 2 triggering, and (**c**) Follower 3 triggering.

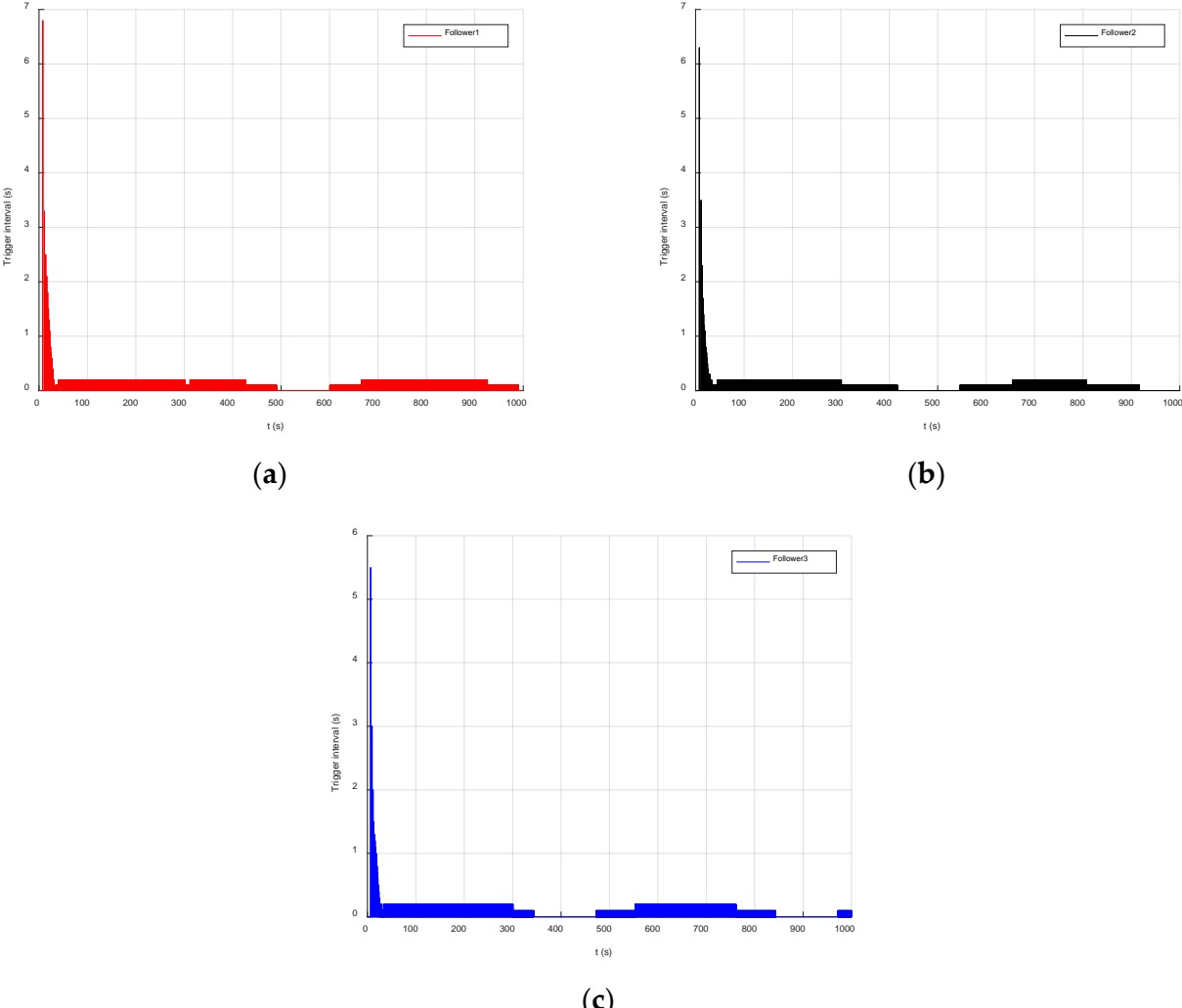

(a)

(b)

(c)

**Figure 12.** The follower event-triggered situation under the algorithm in the literature [13]. (**a**) Follower 1 triggering, (**b**) Follower 2 triggering, and (**c**) Follower 3 triggering.

Table 5 shows the triggering times and rate of the follower AUVs' controller under the event-triggered formation controller. First, the event-triggered times show that the triggering times of the event-triggered function designed in this paper are far less than those of the algorithm in reference [13]. The event-triggered mechanism designed in this paper reduces the traffic by 90%, while the literature [13] only reduces the traffic by 50%. The optimization of the original triggering mechanism is completed by adding an error threshold convention in the thesis which can avoid unnecessary triggering. Second, Figures 11 and 12 show that in the process of a direct course formation, the event-triggered interval of the follower is roughly about 1.5 s, because the ocean current disturbance was added in the simulation. In order to overcome the influence of the ocean current disturbance, the follower needs to update the control input at a certain interval. During the formation process of spiral diving in a three-dimensional space, the average sampling intervals of followers 1–3 were 0.9 s, 0.8 s and 0.7 s, respectively.

## 6. Conclusions

The thesis researches the leader failure in a multi-AUV leader–follower formation. It deals with the problem of leader failure at the decision-making level by using an improved Hungarian algorithm and proposes an improved event-triggered mechanism to solve the problem of the large communication task of the leader AUV during the formation and formation reconfiguration process. The simulation results show that the improved Hungarian algorithm can achieve a formation reconfiguration at a minimum cost, and that it can reduce 85% of the redundant communication under the action of an event-triggered mechanism, which realizes an efficient formation control. The improvement of the formation reconfiguration cost model and time departure mechanism algorithm is still a key point of interest for the future.

Formation reconfiguration control of multiple AUVs is a very complex problem. Despite the above-mentioned research results, in view of the limitations of the talents and space, this paper inevitably has the following shortcomings:

1.  This article uses the full-drive AUV model, which reduces the difficulty of research in the model feedback linearization and controller design. The under-driven AUV model can be followed by related feedback linearization and controller design research.
2.  The formation constraint in this paper is the formation of the horizontal plane, therefore the formation constraint conditions of multiple AUVs in the three-dimensional space need to be studied in depth.

**Author Contributions:** Conceptualization, J.L. and Y.Z.; methodology, J.L.; software, J.L.; validation, J.L. and Y.Z.; formal analysis, J.L.; investigation, J.L. and Y.Z. resources, J.L.; data curation, J.L.; writing—original draft preparation, J.L.; writing—review and editing, J.L. and Y.Z.; visualization, Y.Z. and W.L.; supervision, J.L.; project administration, J.L.; funding acquisition, J.L. All authors have read and agreed to the published version of the manuscript.

**Funding:** This research work was supported by the National Natural Science Foundation of China (Grant No. 51809060/E091002), the National Natural Science Foundation of China (Grant No. 62101156) and the Central University Fund (Grant No. 3072021CFJ0407).

**Data Availability Statement:** Not applicable.

**Conflicts of Interest:** The authors declare no conflict of interest.

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
