# Peer review of "Formation Control of a Multi-Autonomous Underwater Vehicle Event-Triggered Mechanism Based on the Hungarian Algorithm"

_machines, doi:10.3390/machines9120346_

Round 1

Reviewer 1 Report

This paper proposed a novel method to solve the problem of leader failure in multi-AUV leader-follower formations by using Hungarian algorithm. According to the paper, the results show that the redundant communication under the action of event-triggered mechanism is hugely reduced. However, there are several questions worthy to be explained more clearly. Detailed suggestions are as follows:

  1. In the Introduction, the definition of leader failure during the formation control needs to be explained more clearly. Also, the situation of this research requires more details to correspond to the simulation results.
  2. In Section 5 Simulation verification and analysis, the results of formation pay more attention to the leader failure of formation reorganization, however, considering the relationship between communication change and formation control, the explanation of control stability needs an error analysis diagram of 5 DOFs.

Author Response

亲爱的评审员:
我们衷心感谢您仔细阅读đ的一篇有益的评论,以提高这篇论文的质量。我们讨论了所提出的所有重大和次要问题。所作的修改在下文提到,并参考了修改后的手稿的适当段落和章节。

[C'omnent 1]  In the Introduction, the definition of leader failure during the formation control needs to be explained more clearly. Also, the situation of this research requires more details to correspond to the simulation results.
[Answer] Thanks a lot for this comments. In the second paragraph of the introduction, an explanation of the failure of the leader has been added. At the same time, a detailed explanation of the simulation results has been added to the simulation part of the paper.

[C'omnent 2]  In Section 5 Simulation verification and analysis, the results of formation pay more attention to the leader failure of formation reorganization, however, considering the relationship between communication change and formation control, the explanation of control stability needs an error analysis diagram of 5 DOFs.
[Answer] Thank you so much for this comment. In the fifth part of the article, a schematic diagram of five degrees of freedom when formation reconstruction and communication changes are added, and the schematic diagram is  analyzed.

Please see the attachment for the detailed modification part.

Reviewer 2 Report

Formation Control of Multi-AUV Event-triggered Mechanism Based on Hungarian Algorithm
=====================================================================================
> Abstract has a lot of repeatability of "event-triggered mechanism". 
There is a need to provide an acronym to this so that readers have better readability.
Also, there is a need to define acronyms at the start before they are used in the manuscript.
The manuscript has no such 

> The abstract needs to be revised in a better language by stating why the study is done? which research question is answered?
How did you approach the solution? and how do the final results show improvement?

> Some new references in the area of multi-robot systems in the maritime domain needs to be included :
- Singh, Y., Bibuli, M., Zereik, E., Sharma, S., Khan, A. and Sutton, R., 2020. A novel double-layered hybrid multi-robot framework for guidance and navigation of unmanned surface vehicles in a practical maritime environment. 
Journal of Marine Science and Engineering, 8(9), p.624.

> In the decision flow chart provides arrows and direction for flow (whether it is bidirectional).

> Literature review looks bifurcated and there is a need to follow the system in writing the state of the art.
The system should be: 
What are the major problems in the multi-AUV system? how did authors in literature approach the problem?
Which problem is still a challenge? The authors should then provide major contributions of this study as a sub-section. 

> Figure 2 needs revision. Please refer to Fossen's book to revise the figure.

> Variable and terms used in the equations have not been defined, no reference for equations (where they are taken from), plus many variables and symbols
used in the equation have been interchanged in the usage. Refine the sections where equations, symbols, and variables are defined.

> Linearised model is used here while the AUV is a highly non-linear system. Provide reason.

> Provide a section to benchmark the solution with the existing solution in literature or with some experimental solution. Currently,
no benchmarking has been done for the proposed solution. 

> What is the unit of cost? Is it in CPU Time or is it in terms of some other performance indicator?

> Resolution of figures should be improved. Currently, it is very poor.

> How does the proposed solution work in presence of ocean currents, winds, and dynamic obstacles. Provide a detailed section on this with some simulation results.

> A very poor technical writing is found in the discussion section. 
Revise the writing in consultation with some native English speakers to improve readability.

Author Response

亲爱的评审员:

We would 1ike to express our sincere appreciation for your careful reading anđ helpful commentsto improve the quality of this paper. We have addressed all the major and minor issues raised. Theamendments made are mentioned below with reference to appropriate paragraphs and sections ofthe revised manuscript.

[C'omnent 1]   Abstract has a lot of repeatability of "event-triggered mechanism". There is a need to provide an acronym to this so that readers have better readability.Also, there is a need to define acronyms at the start before they are used in the manuscript.The manuscript has no such.

[Answer] Thanks a lot for this comments.Abbreviate the event trigger mechanism in the abstract section of the article, and make sure that all abbreviations are defined in the abstract section.

[C'omnent 2] > The abstract needs to be revised in a better language by stating why the study is done? which research question is answered? How did you approach the solution? and how do the final results show improvement?

[Answer] Thank you so much for this comment. In order to make the article more logical and readable, the reason and background of the research on the problem are added at the beginning of the abstract. The middle part of the abstract explains the methods used and the problems solved, and the end of the abstract explains the results of this research.

[C'omnent 3]   Some new references in the area of multi-robot systems in the maritime domain needs to be included :

- Singh, Y., Bibuli, M., Zereik, E., Sharma, S., Khan, A. and Sutton, R., 2020. A novel double-layered hybrid multi-robot framework for guidance and navigation of unmanned surface vehicles in a practical maritime environment. Journal of Marine Science and Engineering, 8(9), p.624.

[Answer] Thanks a lot for this comments. In order to make the importance of AUV work under water more convincing, this article is quoted.

[C'omnent 4] >  In the decision flow chart provides arrows and direction for flow (whether it is bidirectional).

[Answer] Thank you so much for this comment. In order to reflect the relationship between the decision-making part and the control part in the flowchart, arrow direction is added to the flowchart.

[C'omnent 5]Literature review looks bifurcated and there is a need to follow the system in writing the state of the art.

The system should be:

What are the major problems in the multi-AUV system? how did authors in literature approach the problem?

Which problem is still a challenge? The authors should then provide major contributions of this study as a sub-section.

[Answer] Thanks a lot for this comments. In order to make the introduction more systematic, existing problems and solutions, as well as remaining challenges, are added to the third and fourth paragraphs of the introduction to increase the logic and readability of the article.

[C'omnent 6] > Figure 2 needs revision. Please refer to Fossen's book to revise the figure.

[Answer] Thank you so much for this comment. Figure 2 has been modified.

[C'omnent 7]Variable and terms used in the equations have not been defined, no reference for equations (where they are taken from), plus many variables and symbols used in the equation have been interchanged in the usage. Refine the sections where equations, symbols, and variables are defined.

[Answer] Thanks a lot for this comments.Clarified the source of the formulas in Section 2.2 and Section 3.1 of the article, and added a citation in the references.

[C'omnent 8] > Linearised model is used here while the AUV is a highly non-linear system. Provide reason.

[Answer] Thank you so much for this comment. Section 2.2 of the article performed feedback linearization on the AUV model, so the AUV linearization model is used later, and the detailed process of feedback linearization is added in this section.

[C'omnent 9]Provide a section to benchmark the solution with the existing solution in literature or with some experimental solution. Currently,no benchmarking has been done for the proposed solution.

[Answer] Thanks a lot for this comments.In order to reflect the superiority of the improved method, in the fifth part of the article, the research results of this article are compared with the research results of literature [7]. At the same time, I apologize for not comparing the formation reconstruction algorithm studied in this article with other formation reconstruction algorithms.

[C'omnent 10] > What is the unit of cost? Is it in CPU Time or is it in terms of some other performance indicator?

[Answer] Thank you so much for this comment. The unit of path cost is energy consumed, and the unit of communication propagation loss is decibel. Both units are supplemented in the text.

[C'omnent 11] Resolution of figures should be improved. Currently, it is very poor.

[Answer] Thanks a lot for this comments.In order to make the research results clearer, the resolution of Figure 1, Figure 4, and Figure 9 in the article has been improved.

[C'omnent 12] > How does the proposed solution work in presence of ocean currents, winds, and dynamic obstacles. Provide a detailed section on this with some simulation results.

[Answer] Thank you so much for this comment. I'm very sorry for not explaining how the system will operate in the presence of ocean currents and dynamic obstacles. Due to space limitations and the content of this research deviates from the focus of this article, we will continue to explore this issue in the next article.

[C'omnent 13] > A very poor technical writing is found in the discussion section.

[Answer] Thank you so much for this comment. Some articles in the introduction section have been revised, and some references to other articles have been added.

Please see the attachment for the detailed modification part.

Reviewer 3 Report

This study focuses on formation control of multi-AUV event-triggered mechanism based on Hungarian algorithm. I think the paper fits well the scope of the journal and addresses an important subject. However, a number of revisions are required before the paper can be considered for publication. There are some weak points that have to be strengthened. Below please find more specific comments:

*Please avoid using abbreviations in the title. Many readers may not be familiar with the term “AUV”.

*Please make sure that all the abbreviations are defined in the abstract.

*I suggest starting the abstract with one or two generic sentences that highlight the importance of the problem and studies subject. Starting with the algorithm right away may not seem appropriate to some readers.

*The first paragraph of the introduction section includes quite a few generic phrases, but these phrases are not supported by any relevant references. Please include some supporting references for the statements made.

*I see that some of the relevant studies are discussed in the introduction section. Please check and make sure that all the recent studies relevant to the topic have been captured and acknowledged.

*Page 5 line 156: “The thesis mainly studies” should be replaced with “This manuscript mainly studies”.

*Section 3: Before formally describing the proposed solution algorithm, the authors should create a general discussion regarding the importance of advanced optimization algorithms (e.g., heuristics, metaheuristics) for challenging decision problems. There are many different domains where advanced optimization algorithms have been applied as solution approaches, such as online learning, scheduling, multi-objective optimization, vehicle routing, medicine, data classification, and others (not just multi-AUV operations). The authors should create a discussion that highlights the effectiveness of advanced optimization algorithms in the aforementioned domains. This discussion should be supported by the relevant references, including the following:

  • An online-learning-based evolutionary many-objective algorithm. Information Sciences 2020, 509, pp.1-21.
  • An Adaptive Polyploid Memetic Algorithm for scheduling trucks at a cross-docking terminal. Information Sciences 2021, 565, pp.390-421.
  • AnD: A many-objective evolutionary algorithm with angle-based selection and shift-based density estimation. Information Sciences 2020, 509, pp.400-419.
  • An Optimization Model and Solution Algorithms for the Vehicle Routing Problem with a “Factory-in-a-Box”. IEEE Access 2020, 8, pp.134743-134763.
  • A proposal for distinguishing between bacterial and viral meningitis using genetic programming and decision trees. Soft Computing 2019, 23(22), pp.11775-11791.
  • Hybridized classification algorithms for data classification applications: A review. Egyptian Informatics Journal 2021, 22(2), pp.185-192.

Such a discussion will help improving the quality of the manuscript significantly.

*The presentation of section 4 seems to be adequate.

*Section 5 has a lot of supporting figures and tables. I suggest extending explanations of the results presented in tables and figures where appropriate.

*The conclusions section should expand on limitations of this study and future research needs. I suggest listing the bullet points.

Author Response

亲爱的评审员:

We would 1ike to express our sincere appreciation for your careful reading anđ helpful commentsto improve the quality of this paper. We have addressed all the major and minor issues raised. Theamendments made are mentioned below with reference to appropriate paragraphs and sections ofthe revised manuscript.

[C'omnent 1] Please avoid using abbreviations in the title. Many readers may not be familiar with the term “AUV”.

[Answer] Thanks a lot for this comments. The abbreviation in the title has been changed to the full name

[C'omnent 2] *Please make sure that all the abbreviations are defined in the abstract.

[Answer] Thank you so much for this comment. It has been ensured that all abbreviations are defined in the abstract.

[C'omnent 3]

*I suggest starting the abstract with one or two generic sentences that highlight the importance of the problem and studies subject. Starting with the algorithm right away may not seem appropriate to some readers.

[Answer] Thank you so much for this comment. At the beginning of the abstract, the background and reasons for the research content of this article are added, and the logic and readability of the article are increased.

[C'omnent 4]The first paragraph of the introduction section includes quite a few generic phrases, but these phrases are not supported by any relevant references. Please include some supporting references for the statements made.

[Answer] Thanks a lot for this comments. The first paragraph of the introduction was revised, and references were added to support common phrases that appeared.

[C'omnent 5] *I see that some of the relevant studies are discussed in the introduction section. Please check and make sure that all the recent studies relevant to the topic have been captured and acknowledged.

[Answer] Thank you so much for this comment. The latest research related to this topic has been checked and confirmed.

[C'omnent 6] Page 5 line 156: “The thesis mainly studies” should be replaced with “This manuscript mainly studies”.

[Answer] Thank you so much for this comment. This part has been revised in accordance with the comments.

[C'omnent 7]Section 3: Before formally describing the proposed solution algorithm, the authors should create a general discussion regarding the importance of advanced optimization algorithms (e.g., heuristics, metaheuristics) for challenging decision problems. There are many different domains where advanced optimization algorithms have been applied as solution approaches, such as online learning, scheduling, multi-objective optimization, vehicle routing, medicine, data classification, and others (not just multi-AUV operations). The authors should create a discussion that highlights the effectiveness of advanced optimization algorithms in the aforementioned domains. This discussion should be supported by the relevant references, including the following:

  • An online-learning-based evolutionary many-objective algorithm. Information Sciences 2020, 509, pp.1-21.
  • An Adaptive Polyploid Memetic Algorithm for scheduling trucks at a cross-docking terminal. Information Sciences 2021, 565, pp.390-421.
  • AnD: A many-objective evolutionary algorithm with angle-based selection and shift-based density estimation. Information Sciences 2020, 509, pp.400-419.
  • An Optimization Model and Solution Algorithms for the Vehicle Routing Problem with a “Factory-in-a-Box”. IEEE Access 2020, 8, pp.134743-134763.
  • A proposal for distinguishing between bacterial and viral meningitis using genetic programming and decision trees. Soft Computing 2019, 23(22), pp.11775-11791.
  • Hybridized classification algorithms for data classification applications: A review. Egyptian Informatics Journal 2021, 22(2), pp.185-192.

Such a discussion will help improving the quality of the manuscript significantly.

[Answer] Thanks a lot for this comments. In the third part of the article, a discussion of advanced optimization algorithms to solve decision-making problems is established, and the following documents are cited.

[C'omnent 8] Section 5 has a lot of supporting figures and tables. I suggest extending explanations of the results presented in tables and figures where appropriate.

[Answer] Thank you so much for this comment. Added the explanation of some pictures and tables in the simulation part.

[C'omnent 9] The conclusions section should expand on limitations of this study and future research needs. I suggest listing the bullet points.

[Answer] Thank you so much for this comment. In the conclusion part, the deficiencies of this article's research and prospects for the future are added.

Please see the attachment for the detailed modification part.

Round 2

Reviewer 2 Report

The required comments have been addressed by the authors

Reviewer 3 Report

The authors took seriously my previous comments and made the required revisions in the manuscript. The quality and presentation of the manuscript have been improved. Therefore, I recommend acceptance.